# Exploring One Million Machine Learning Pipelines: A Benchmarking Study

**Edesio Alcobaça**[1]   **André C. P. L. F. De Carvalho**[1]

[1]University of São Paulo

**Abstract**  Machine learning solutions are largely affected by the values of the hyperparameters of their algorithms. This has motivated a large number of recent research projects on hyperparameter tuning, with the proposal of several, and highly diverse, tuning approaches. Rather than proposing a new approach or identifying the most effective hyperparameter tuning approach, this paper looks for good machine learning solutions by exploring machine learning pipelines. For such, it benchmarks pipelines focusing on the interaction between feature preprocessing techniques and classification models. The study evaluates the effectiveness of pipeline combinations, identifying high-performing and underperforming combinations. Additionally, it provides meta-knowledge datasets without any optimization selection bias to foster research contributions in meta-learning, accelerating the development of meta-models. The findings provide insights into the most effective preprocessing and modeling combination, guiding practitioners and researchers in their selection processes.

## 1 Introduction

Despite being seen as a transformative technology across various domains [1–3], the performance obtained by machine learning models strongly depends on how effective are the stages of data preparation, model selection, and evaluation. This process is often structured as a machine learning pipeline. This sequential workflow combines preprocessing stages and model training in a unified way [4].

Despite the importance of pipelines, their design and performance across different problems remain a topic of ongoing research. Benchmarking the effectiveness of various pipelines, particularly by exploring different combinations of preprocessing techniques and modeling algorithms, is crucial to understanding their strengths and limitations [5–8]. This study aims to benchmark machine learning pipelines across diverse classification and preprocessing algorithms.

The main objectives of this research are as follows: (i) to evaluate machine learning models interaction with different feature preprocessing techniques; (ii) to identify which combinations of preprocessing stages and models are most effective and which ones perform poorly; (iii) to generate valuable meta-knowledge datasets, without any optimization selection bias, to foster research in meta-learning and automated machine learning (AutoML).

This paper contributes by offering insights into the performance of various machine learning pipelines across a wide range of classification tasks. The findings aim to guide practitioners and researchers in selecting effective preprocessing and modeling techniques. Furthermore, the pipeline runs generated from this study were properly collected in order to serve as an off-the-shelf meta-knowledge for meta-learning approaches, enhancing the development of AutoML systems [9].

The structure of this paper is as follows: Section 2 reviews the existing literature on machine learning pipelines and hyperparameter tuning. Section 3 describes the experimental design, including the datasets, models, and preprocessing techniques used. In Section 4, we present the results of the benchmarking experiment and discuss the findings and their implications. Finally, Section 5 concludes the paper with a summary of key insights and suggestions for future work.

## 2 Background

The design and optimization of machine learning pipelines have been the focus of extensive research, encompassing both hyperparameter tuning of individual components and integrating preprocessing and modeling stages [10–12]. These studies have significantly advanced the field by providing insights into learning, default hyperparameters [13], hyperparameter optimization initialization [14, 15] and tunability [16–18].

Several studies have examined the impact of hyperparameter tuning on the performance of individual classifiers. Probst *et al.* [18] explored tuning strategies for Random Forest models, providing insights into key hyperparameters that affect their predictive accuracy. Similarly, Mantovani *et al.* [19] conducted studies on hyperparameter tuning for Support Vector Machines (SVM), emphasizing the effectiveness of random search for identifying optimal configurations. Futhermore, Bentéjac *et al.* [20] provided a comparative analysis of gradient boosting algorithms, highlighting the impact of parameter tuning on model performance. Beyond algorithm selection, portfolio-based hyperparameter optimization has been a key focus. For instance, Bardenet *et al.* [21] introduced collaborative hyperparameter tuning, while Bergstra & Bengio [10] demonstrated the efficacy of random search over grid search. Bayesian optimization techniques, as discussed by Wu *et al.* [22] and Snoek *et al.* [23], offer probabilistic models to efficiently explore the hyperparameter space. Evolutionary approaches, such as hierarchical ant colony optimization by Costa & Rodrigues [24], have also shown promise for simultaneous classifier selection and hyperparameter tuning.

Preprocessing is another key component of machine learning pipelines, as demonstrated by García *et al.* [25], who explored various preprocessing methods for large datasets. Bommert *et al.* [26] benchmarked filter methods for feature selection, showcasing their effectiveness in improving model performance. Similarly, Alcobaça *et al.* [27] studied the impact of dimensionality reduction techniques in mitigating the curse of dimensionality in high-dimensional datasets. Furthermore, Jäger *et al.* [28] examined data imputation methods to improve data quality. These studies highlight the importance of systematically evaluating preprocessing methods and their interactions with modeling algorithms.

Integrating preprocessing and modeling stages into a unified pipeline has been a focus of AutoML frameworks [6, 7]. Feurer *et al.* [4] introduced Auto-sklearn 2.0, which leverages meta-learning for hands-free optimization of machine learning pipelines. Olson & Moore [29] developed TPOT, a tree-based optimization tool, while Erickson *et al.* [30] proposed AutoGluon-Tabular, emphasizing robust and accurate AutoML for structured data. Other notable contributions include Auto-WEKA, FLAML, and H2O AutoML [31–33]. Benchmarking studies are crucial in evaluating and comparing machine learning frameworks and methods. Zöller & Huber [6] surveyed automated machine learning frameworks, and Olson *et al.* [5] introduced PMLB, a benchmark suite for machine learning evaluation. Gijsbers *et al.* [7] and Eldeeb *et al.* [8] provided comprehensive benchmarks for AutoML frameworks, facilitating objective comparisons.

While existing studies have extensively explored hyperparameter tuning and pipeline optimization, this research differs in key aspects. Rather than focusing on identifying the most effective tuning approach, we aim to evaluate the interactions between classifiers and preprocessors within pipelines. By employing a random search strategy, we eliminate biases introduced by model-based optimization methods, enabling an unbiased evaluation of pipeline compositions. Furthermore, this study aims to generate meta-knowledge to support meta-learning approaches for pipeline selection and optimization, paving the way for more effective AutoML systems.

## 3 Methodology

This section provides a detailed description of the methodological framework adopted in this study. It outlines the dataset selection criteria, the design of the configuration space for machine learning pipelines, and the experimental setup.

### 3.1 Datasets

For this study, we collected 211 real-world datasets from diverse domains to ensure comprehensive coverage of various classification problems. The datasets were collected using the OpenML [34] and were selected based on curated collection from previous AutoML work [4, 7]. The selection criteria required datasets to have between 500 and 1,000,000 samples of classification tasks. Datasets OpenML IDs and meta-data can be found in Table 4 in the Appendix.

### 3.2 Configuration Space

The configuration space defines the range of possible pipeline designs, encompassing key components where the search operates. In this study, the configuration space is structured as a tree-based search space, following [35], with three main components: modeling, feature preprocessing, and data preprocessing.

Figure 1 illustrates the hierarchical structure of the configuration space. The root node represents the pipeline, branching into the three primary components—each component further branches into its respective hyperparameters, with dimensions defining these parameters' potential values.

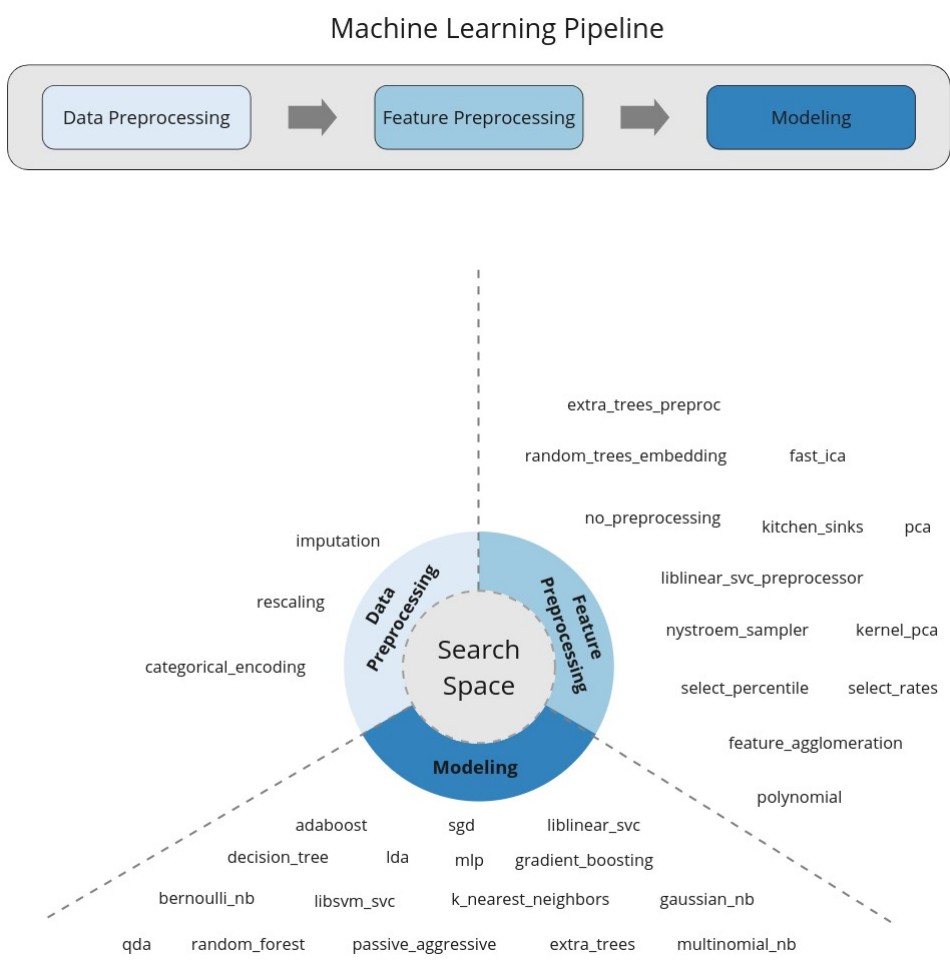

Figure 1: Main stages the machine learning pipeline considered. The first levels of search space with available algorithms are shown.

We used 16 classification algorithms, encompassing a broad spectrum of algorithmic categories. These included distance-based methods (e.g., k-nearest neighbors), linear models (e.g., stochastic

gradient descent), neural networks (e.g., multilayer perceptron), kernel-based approaches (e.g., support vector machines), probabilistic methods (e.g., naive Bayes), tree-based algorithms (e.g., decision tree, extra trees), and ensemble-based (e.g., gradient boosting, random forest).

We employed 13 feature preprocessing techniques representing distinct strategies. These included dimensionality reduction methods (e.g., principal component analysis, independent component analysis), feature selection techniques (e.g., selecting percentiles based on feature statistics, rate-based), feature generation approaches (e.g., polynomial feature generation), embedding-based methods (e.g., random tree embeddings), model-based feature selection (e.g., support vector coefficients, extra-tree feature selection), and a no-preprocessing option. Data preprocessing was applied when required to ensure compatibility with the scikit-learn algorithms. We used one-hot encoding to transform categorical features, data imputation to fill in missing values, and feature rescaling to standardize or normalize feature distributions. Tables 1, 2, and 3 in the Appendix provide detailed descriptions of the configuration space dimensions for modeling, feature preprocessing, and data preprocessing, respectively. These tables outline the possible values, hyperparameter categories, and indicate whether transformations such as $log_{10}$ scaling are applied.

### 3.3 Experimental Setup

The datasets were split into training, validation, and testing. We used the holdout approach for the train-test split, allocating 25% of the data for testing. The training set was split using a 10-fold cross-validation approach to assess validation performance. Thus, validation performance was assessed using cross-validation, while test performance by holdout.

To select the pipeline configuration, we used random search as the search strategy, ensuring a broad and unbiased exploration of the configuration space. This approach minimizes the selection bias once there are no assumptions when selecting a new sample. Thus, we can search for configurations without prioritizing regions. Therefore, it is fair to assume that configuration regions were randomly visited equally. Each of the 211 datasets was evaluated on a sample of 500 different pipeline configurations. It resulted in $500 \times 211 \times 10 = 1,055,000$ pipeline evaluations.

We evaluated performance using multiple metrics, including accuracy, balanced accuracy, F1-macro, F1-weighted, precision-macro, precision-weighted, recall-macro, and recall-weighted. Each metric was computed for the training, validation, and test sets, providing a comprehensive assessment of each pipeline run. The experiments were implemented using Scikit-learn algorithms orchestrated by Auto-sklearn [35, 36], with a limit of 600 seconds per pipeline run and 24 hours per dataset. The experiments run on a Debian Linux system, using Intel Xeon E5-2680v2 and 128 GB DDR3 RAM. Each pipeline execution was constrained to a maximum memory usage of 10 GB and a single computational core. To promote reproducibility and facilitate further research, all experimental code and analysis are available on GitHub [1]. Furthermore, we provide a comprehensive dataset containing the pipeline configurations, execution times, and performance metrics obtained during the experiments [2]. This dataset is a valuable resource for enhancing meta-knowledge in AutoML systems and supports the research of more robust and efficient machine learning pipelines.

## 4 Experimental Results

This section presents and explores the main experiments carried out in this study. The analysis of the experimental results is divided into three main aspects: feature preprocessor performance, modeling (classifier) performance, and pipeline performance. While pipeline performance is discussed in this section, the same analysis for feature preprocessing and classifier performance is left in Appendix A and Appendix B, respectively. For this reported analysis, we focused on the F1-weighted score. We chose this metric because it accounts for class imbalance, ensuring that the performance of

---

[1]GitHub Repository: https://github.com/ealcobaca/exploring-machine-learning-pipelines
[2]Pipeline database can be found at figshare.

minority classes was adequately represented and reducing the potential bias toward the majority class.

For each pipeline and dataset, we aggregated the test performance across 10 runs using statistical measures: mean, median, standard deviation, and maximum values. This allowed us to evaluate the overall consistency and variability in performance. To visualize the distribution of results, we employed boxplots. Furthermore, for each aggregation, we computed the ranking of each method, where values closer to 1 indicate superior performance. In addition to ranking, we tracked how often each method won, tied, or lost. In this section, we concentrate on maximum and standard deviation aggregations, while the analysis of mean and median is presented in Appendix D.

To assess the statistical significance of differences between methods, we applied the Friedman-Nemenyi test. This non-parametric test enabled us to determine whether there were statistically significant differences, helping to identify which algorithms performed best [37].

We also measured each method's computational efficiency by tracking the time spent during pipeline execution. This metric reflects the processing time required for each pipeline, including the feature preprocessing and classifier stages. Lastly, we evaluated each pipeline's success rate, considering timeouts and memory usage factors. We documented the rates of pipeline failures, either due to exceeding the time limit or running out of memory, which is critical for understanding the scalability and reliability of each approach in practical scenarios.

## 4.1 Performance obtained by the Pipelines

Figure 2 presents the aggregated maximum and standard deviation F1 performance of pipelines by dataset for each combination of classifier and feature preprocessor for the 20 outperforming combinations. The boxplot of the aggregated maximum shows that the combination of Gradient Boosting with Polynomial Features achieved the highest performance, followed by Gradient Boosting with Feature Agglomeration, AdaBoost with Feature Agglomeration, and Extra Trees without preprocessing. Tree-based ensemble classifiers, combined with feature generation approaches like Polynomial Features and dimensionality reduction methods such as Feature Agglomeration, had highest performance.

The ranking analysis indicated that Extra Trees without preprocessing achieved the best rank score. In terms of wins, AdaBoost with Polynomial Features was the top combination with 12 wins, followed by Gradient Boosting with Polynomial Features, which had 11 wins.

The standard deviation analysis showed that SVM combined with Select Percentile, SGD with Select Rates, and SVM with Select Rate exhibited the highest variability in performance across datasets. Additionally, in the win/tie analysis, the combination of MLP with Extra Trees Preprocessor got 15 wins.

Figure 3 presents the aggregated maximum and standard deviation F1 performance of pipelines by dataset for the 20 underperforming combinations. The boxplot of the aggregated maximum F1 scores indicates that the combination of Naive Bayes with kernel transformation exhibited the poorest performance, followed by SGD with Fast ICA and SGD with Kitchen Sinks. These combinations involve linear modeling approaches, which are inherently simple and require the feature space to be linearly separable to achieve optimal performance. The kernel transformation used in preprocessing may require more intensive hyperparameter optimization to effectively transform the feature space, a process that was limited in this study, likely contributing to the observed poor performance.

The ranking analysis confirmed these results, consistently identifying the three aforementioned combinations as the lowest-performing pipelines, all with no wins in the ranking. Only the combination of SGD with Kitchen Sinks achieved a tie in one instance, while the others resulted exclusively in losses.

The standard deviation analysis revealed that the combination of Decision Tree with Select Rates exhibited very low variability in performance. This behavior can be attributed to the double

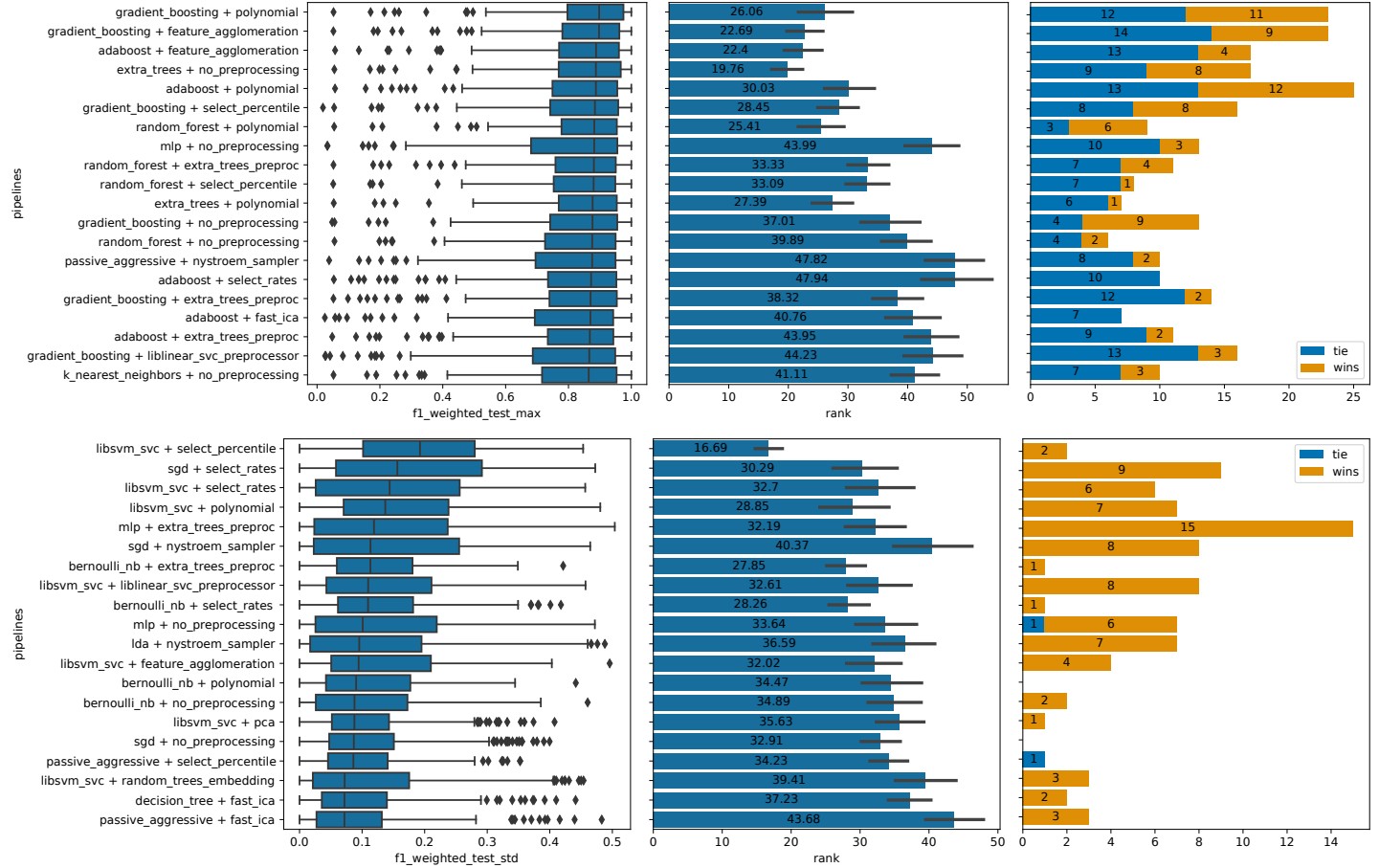

Figure 2: Aggregated F1 by dataset for each machine learning pipeline. From top to bottom, the figure shows F1 performance aggregated using maximum and standard deviation. From left to right, the plots display the performance boxplots, the ranking barplot, and the wins and ties barplot for each pipeline. All plots are ordered by boxplot mean. Only the worst 20 (excepting Kernel PCA) were selected.

filtering effect applied by both the preprocessing method and the classifier itself. Linear models, such as Naive Bayes and SGD-based pipelines, also showed low standard deviation, indicating consistent performance. However, these combinations consistently ranked among the lowest in terms of F1 scores, with poor overall performance.

The findings show that combinations with low standard deviation in performance do not necessarily imply high-performing pipelines. While low variability indicates consistent results across datasets, this consistency can also reflect uniformly poor performance, as observed with linear models and pipelines that applied excessive filtering.

Figure 4 presents the results of the Friedman statistical test and Nemenyi post-hoc analysis for pipeline combinations of preprocessors and classifiers. Extra Trees with no preprocessing, Gradient Boosting with Feature Agglomeration, and AdaBoost with Feature Agglomeration are tied at the top without significant differences. Gradient Boost with Polynomial Feature and Adaboost with Polynomial Feature come next, being statistically different.

Figure 5 presents the analysis of pipeline training time. The slowest pipeline combinations were Gradient Boosting with Polynomial Features, Random Forest with Polynomial Features, and Multilayer Perceptron with Random Tree Embeddings. However, in the ranking score analysis, RF

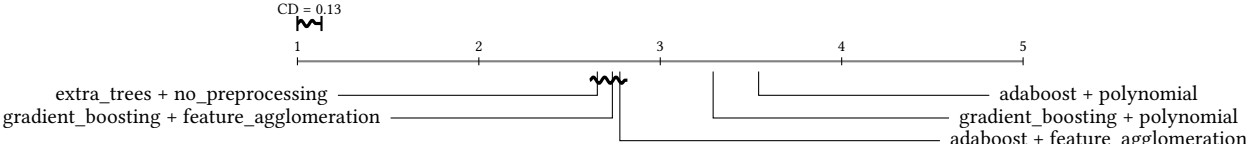

Figure 3: Aggregated F1 by dataset for each for each machine learning pipeline. From top to bottom, the figure shows F1 performance aggregated using maximum and standard deviation. From left to right, the plots display the performance boxplots, the ranking barplot, and the wins and ties barplot for each pipeline. All plots are ordered by boxplot mean. Only the top 20 worst minus Kernel PCA, sorted by boxplot mean, were selected.

Figure 4: Friedman statistical test and Nemenyi post-hoc analysis for machine learning pipelines ($\alpha = 0.05$). The diagram illustrates statistically significant differences among methods, where groups of methods not connected by a line are significantly different. Only the top 5 best, sorted by boxplot median, were selected.

with Fast ICA achieved the highest score. The fastest pipelines were generated using combinations with Naive Bayes classifiers.

Figure 6 shows the percentage of successful runs, time-out errors, and memory-out errors for each pipeline composition. Pipelines with Kernel PCA had a 100% memory-out rate due to the high memory resource needed. Polynomial Feature combinations also exhibited memory-out and time-out errors, ranging from 13% to 21% and 5% to 16%, respectively.

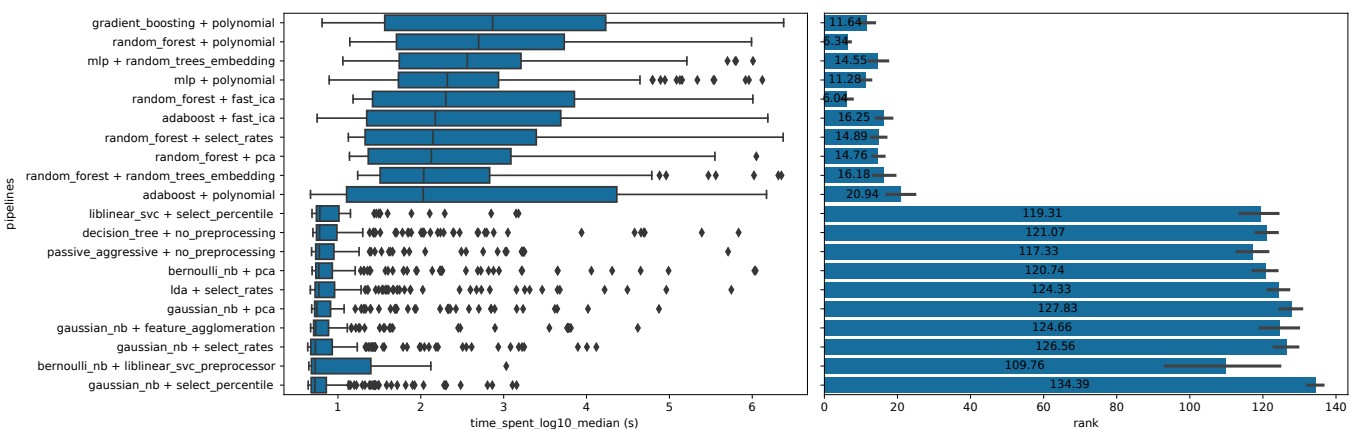

Figure 5: Pipeline training time analysis aggregated by dataset for each pipeline. The left side displays a boxplot of the time spent in seconds after log10 scale, while the right side shows the time-spent rankings. Methods are ordered by the mean time from the boxplot. Only the top 10 outperforming and 10 underperforming pipeline combinations, sorted by f1 aggregate mean, were considered.

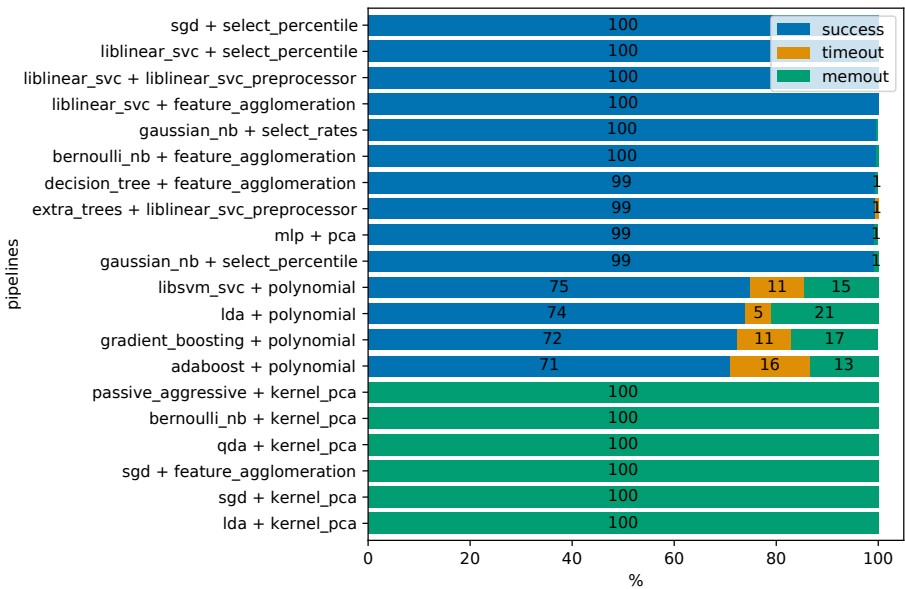

Figure 6: Barplot showing the percentage of successful runs, time-out errors, and memory-out errors for each machine learning pipeline. Only the top 10 outperforming and 10 underperforming pipeline combinations, sorted by f1 aggregate mean, were considered.

## 4.2 Key Insights

The analysis of preprocessing methods revealed that No Preprocessing, Feature Agglomeration, Polynomial Features, and Select Percentile were the most effective in maintaining or improving pipeline performance. These approaches likely preserve or enhance relevant feature structures for subsequent classification. In contrast, kernel PCA and Random Kitchen Sinks demonstrated limited success, possibly due to their sensitivity to hyperparameter tuning. See complete results in Appendix A

The efficacy of No Preprocessing aligns with the restriction of search space of Auto-Sklearn 2.0, which removed feature preprocessing [4]. Furthermore, some empirical studies reveal that feature preprocessing does not always enhance performance, suggesting that in some cases, preprocessing may be unnecessary [38]. In contrast, other studies suggested that processing can reduce the computational training time, which helps to model very large datasets [39, 40].

Ensemble tree-based algorithms, which yielded favorable results in this study, inherently perform feature selection during model construction. Thus, preprocessing methods aimed at reducing feature set size may be redundant for these algorithms. Conversely, feature preprocessing has proven beneficial for algorithms lacking inherent feature selection mechanisms, such as MLP, k-NN, and SVM [41]. Furthermore, Polynomial Features, which generate new features by considering polynomial combinations of existing ones, have been critical to certain AutoML systems, aiding them in achieving high performance [29, 42].

Among classifiers, ensemble-based models such as Gradient Boosting, AdaBoost, Extra Trees, and Random Forest consistently demonstrated strong performance. These models combine multiple base learners to improve predictive performance and handle complex dataset. Conversely, Naive Bayes exhibited poor results, likely due to its simplifying assumption of feature independence, which limits its effectiveness for datasets with complex feature interactions. See complete results in Appendix B

The effectiveness of ensemble models has been extensively documented in the literature, including research in AutoML [29, 35, 43–45]. Ensemble methods typically outperform single models by leveraging the strengths of diverse learners, thereby reducing variance and bias to achieve superior predictive performance [43]. Notably, pipelines incorporating RF not only demonstrated good performance but also exhibited the lowest standard deviation, indicating consistent results across diverse datasets. RF stands out for its robustness and minimal hyperparameter tuning requirements, distinguishing it from more tuning-sensitive algorithms like SVM and MLP [13, 46].

In contrast, kernel-based models, such as SVM, which utilize kernel transformations to map data into higher-dimensional spaces, often require extensive hyperparameter tuning to achieve optimal performance [13, 18]. The relatively poor performance of SVM (and kernel-based preprocessing methods) observed in this study may be attributed to the limited hyperparameter optimization conducted during experimentation.

The combination of ensemble classifiers with selected preprocessing methods was frequently among the top-performing pipeline configurations. Notably, the combination of Extra Trees with No Preprocessing achieved the best results, underscoring the strength of tree-based methods without the added complexity of feature transformations. Additionally, pipelines combining AdaBoost or Gradient Boosting with Feature Agglomeration performed well, likely benefiting from the synergy between these ensemble methods and feature reduction techniques.

Pipelines involving Naive Bayes consistently underperformed, regardless of the preprocessing method used. This was particularly evident when paired with preprocessing approaches such as Select Percentile or kernel-based transformations. These combinations likely failed due to Naive Bayes' limited capacity to model intricate feature dependencies, compounded by the effects of preprocessing methods that alter feature distributions in ways that the classifier could not leverage effectively.

## 4.3 Limitations

Despite this study's comprehensive design, some limitations must be acknowledged. The configuration space, although extensive, can not capture all possible preprocessing and modeling strategies available in the literature. Thus, emerging techniques not currently available in Scikit-learn, such as XGBoost, LightGBM or CatBoost, may outperform on certain datasets. Moreover, we did not use a semantic constraint in the pipeline generation to avoid implausible or suboptimal configurations. Approaches like that can avoid spurious configurations as presented by [47].

Hyperparameter optimization was not exhaustively explored, as predefined ranges were used instead. While this approach relied on Auto-sklearn's configuration space, which has demonstrated efficacy in previous AutoML competitions [4], it may not fully reflect the potential of some algorithms. Moreover, although Random Search is suitable for unbiased sampling, which is our objective in this work, as we aim to provide reusable meta-knowledge for meta-learning systems, the absence of fine-grained hyperparameter optimization likely penalizes tuning-sensitive models, such as SVC, MLP, and preprocessing kernel-based approaches.

The experiments were performed under resource constraints, including the use of a single computational core, a 10 GB memory limit, and a 600-second maximum runtime per pipeline. These limitations likely contributed to failures, particularly for resource-intensive methods such as Kernel PCA and SVM. Relaxing these constraints could alter the outcomes. Lastly, some pipeline configurations, notably those using Kernel PCA and Polynomial Features, experienced frequent memory or time-out errors. These errors likely impacted the observed performance of these methods.

## 5 Conclusions

This study benchmarks machine learning pipelines by systematically analyzing the interaction between classifiers and feature preprocessing techniques across diverse classification tasks. Unlike previous research focusing on individual hyperparameter tuning strategies or optimization approaches, this work evaluates pipeline compositions without introducing exploration assumptions from specific tuning methods. By employing Random Search, we provide an assumption-free exploration of pipeline configurations, generating meta-knowledge that can feed meta-learning approaches for pipeline selection and optimization.

Key findings highlight the effectiveness of techniques like No Preprocessing, Feature Agglomeration, Polynomial Features, and Select Percentile in combination with ensemble classifiers, such as Gradient Boosting, Adaboost and Extra Trees. These classifiers performed well with minimal hyperparameter tuning. In contrast, Kernel PCA, Random Kitchen Sinks, and classifiers like Naive Bayes and SVM underperformed. Future work can leverage the meta-data and insights discovered in this study to enhance the development of AutoML systems by identifying robust pipeline configurations that require minimal tuning. Additionally, this analysis could be extended to include regression and unsupervised learning tasks.

**Acknowledgements**. Research carried out using the computational resources of the Center for Mathematical Sciences Applied to Industry (CeMEAI) funded by FAPESP (grant 2013/07375-0) and FAPESP (grant 2018/14819-5). This study was financed in part by the Coordenação de Aperfeiçoamento de Pessoal de Nível Superior - Brasil (CAPES) - Finance Code 001.

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

## A  Feature Preprocessors Performance

Figure 7 presents the aggregated maximum and standard deviation of F1 performance for pipelines across datasets, grouped by feature preprocessing methods.

The boxplot of the aggregated maximum shows that pipelines without preprocessing achieved the highest performance, followed by Feature Agglomeration and Select Percentile. Kernel PCA and Kitchen Sinks had the lowest performance, with Kernel PCA failing to produce any successful runs. This issue is discussed in Section 4.3.

The ranking results identified No Preprocessing, Polynomial Features, and Feature Agglomeration as the top three methods. In terms of win/tie scores, Polynomial Features achieved 42 wins, followed by No Preprocessing with 28, while Feature Agglomeration and Select Percentile tied with 22 wins each.

The standard deviation analysis showed that Kitchen Sinks, Nystroem Sampler, and Polynomial Features had the highest variation in performance distribution. This pattern was consistent with ranking and win/tie analysis. Both Kitchen Sinks and Nystroem Sampler transform the feature space using kernel approximation techniques.

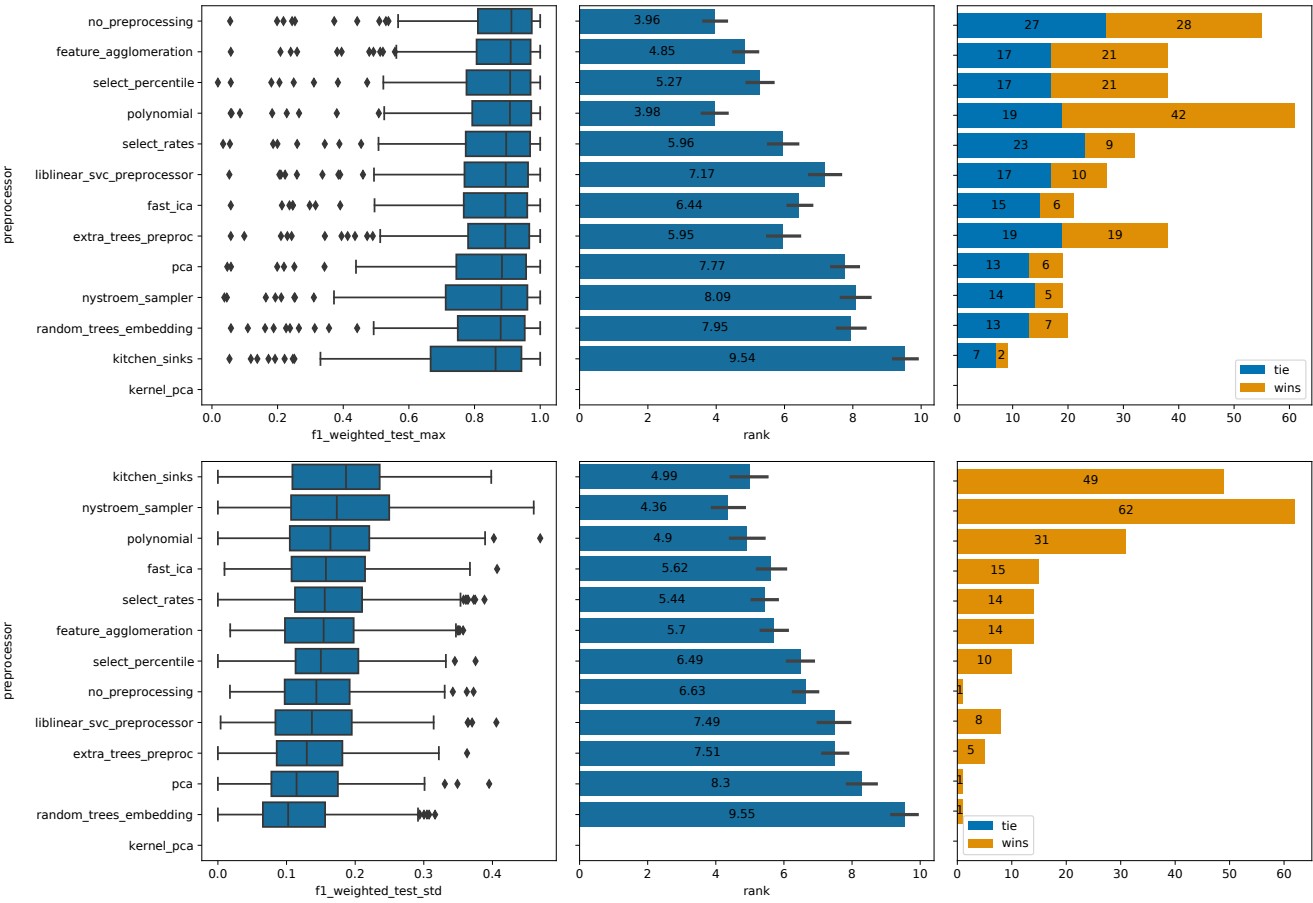

Figure 7: Aggregated F1-score by dataset for each for each feature preprocessor. From top to bottom, the figure shows F1 performance aggregated using maximum and standard deviation values. From left to right, the plots display the performance boxplots, the ranking barplot, and the wins and ties barplot for each method. All plots are ordered by boxplot mean.

Figure 8 presents the Friedman statistical test and Nemenyi post-hoc analysis for feature preprocessing methods. Pipelines without preprocessing demonstrated statistically superior performance

compared to other methods. Feature Agglomeration, Polynomial Features, and Select Percentile followed with no significant differences among them. Excluding Kernel PCA, which did not produce any successful runs, Kitchen Sinks had the lowest performance.

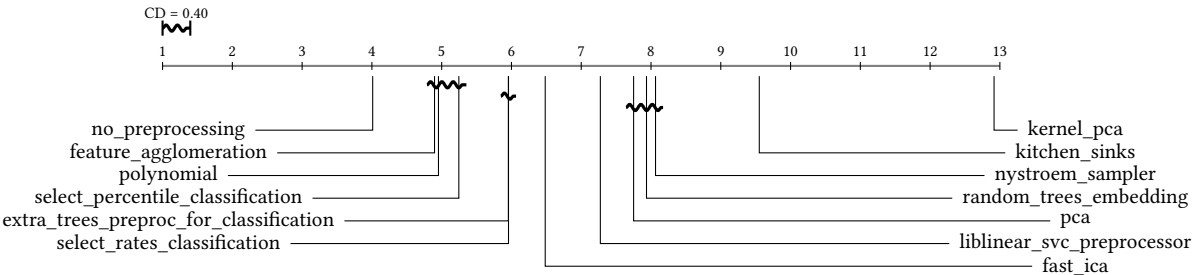

Figure 8: Friedman statistical test and Nemenyi post-hoc analysis for feature preprocessing methods ($\alpha = 0.05$). The diagram illustrates statistically significant differences among methods, where groups of methods not connected by a line are significantly different.

Figure 9 presents the analysis of pipeline training time. The fastest methods were no preprocessing, followed by PCA. Nystroem Sampler and Kitchen Sinks, both kernel transformation methods, were the slowest. However, when considering the ranking score, Fast ICA emerged as the slowest method.

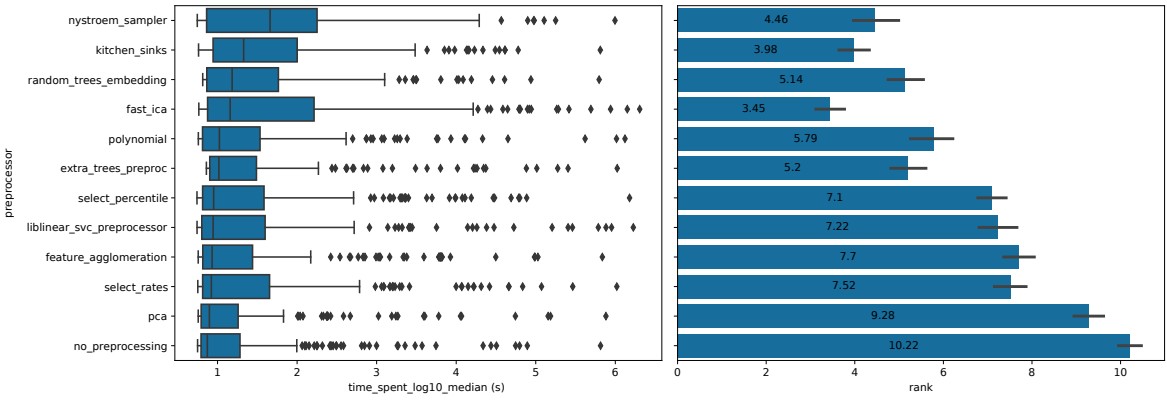

Figure 9: Pipeline training time analysis aggregated by dataset for each feature preprocessor using median. The left side displays a boxplot of the time spent, while the right side shows the time-spent rankings. Methods are ordered by the mean time from the boxplot.

Finally, Figure 10 shows the percentage of successful runs, time-out errors, and memory-out errors for each feature preprocessor. Kernel PCA failed in all runs due to memory-out errors, as all pipelines using this method exceeded the 10 GB memory limit. Polynomial feature generation also exhibited a high memory-out rate of 14% and a time-out rate of 5%. This behavior may be attributed to the algorithm's feature-combination process, which can result in a very high-dimensional feature space when applied to datasets with a large number of features.

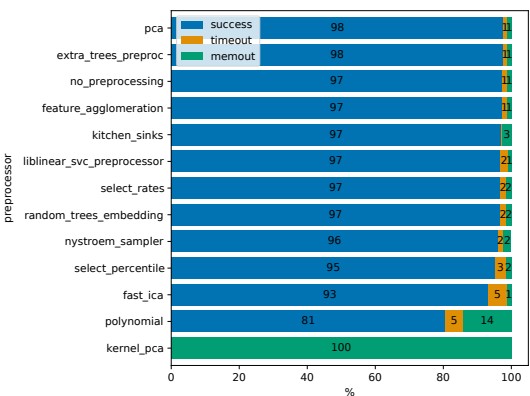

Figure 10: Barplot showing the percentage of successful runs, time-out errors, and memory-out errors for each feature preprocessor method.

## B Classifiers Performance

Figure 11 presents the aggregated maximum and standard deviation F1 performance of pipelines by dataset for each classifier.

The boxplot of the aggregated maximum shows that Extra Trees achieved the highest performance, followed by AdaBoost, Gradient Boosting, and Random Forest (RF). These are tree-based ensemble algorithms with different ensemble approaches: RF uses bagging, training multiple decision trees on bootstrapped data samples and aggregating their predictions, while Gradient Boosting uses boosting, training trees sequentially to correct errors made by previous trees.

Bernoulli, Multinomial, and Gaussian Naive Bayes showed the lowest performance. The Naive Bayes algorithm assumes feature independence, which may not hold for many real datasets. The ranking analysis confirmed the top four classifiers, with Gradient Boosting ranking highest, followed by Extra Trees, AdaBoost, and RF.

The standard deviation analysis showed that SVM, Stochastic Gradient Descent (SGD), and Multilayer Perceptron (MLP) had the highest variability in performance across datasets. The SVM model includes radial basis function, sigmoid, or polynomial kernel transformations, which can be sensitive to hyperparameter settings and dataset characteristics. This trend was consistent with their ranking and win/tie performance. In contrast, RF demonstrated low variance, followed by Multinomial Naive Bayes and k-Nearest Neighbor (k-NN).

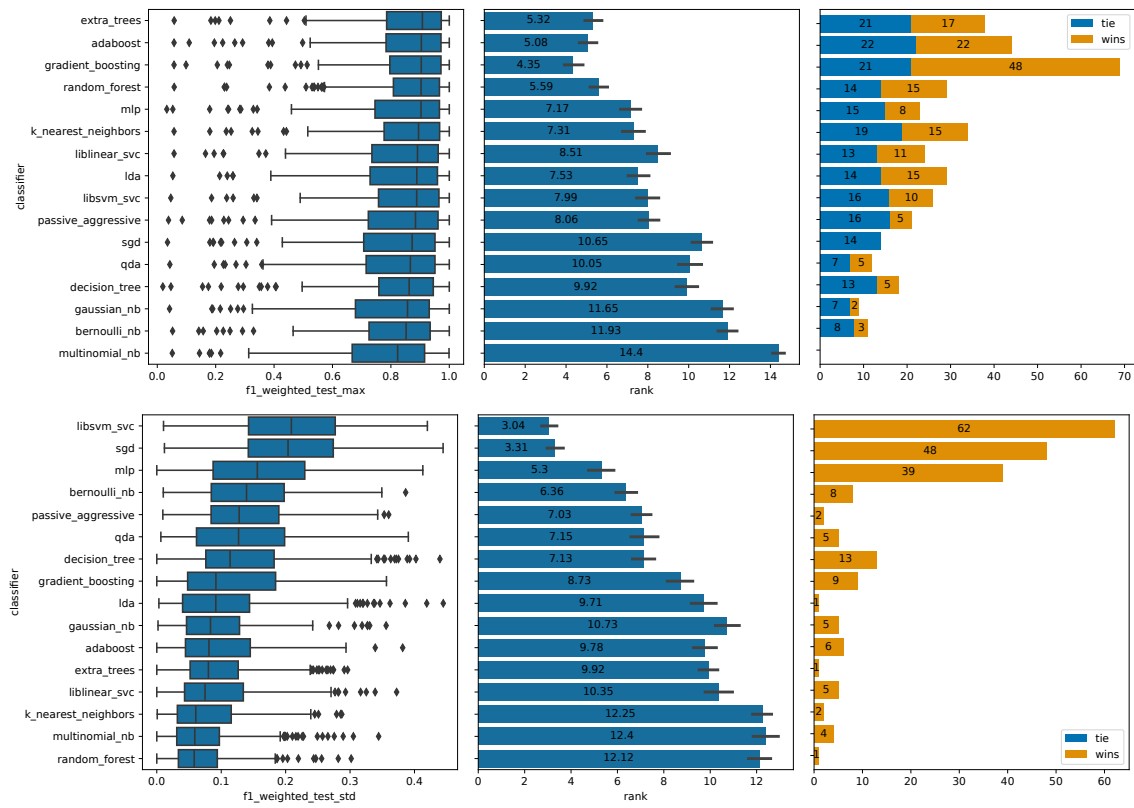

Figure 11: Aggregated F1 by dataset for each for each classifier algorithm. From top to bottom, the figure shows F1 performance aggregated using maximum and standard deviation values. From left to right, the plots display the performance boxplots, the ranking barplot, and the wins and ties barplot for each classifier. All plots are ordered by boxplot mean.

Figure 12 presents the results of the Friedman statistical test and Nemenyi post-hoc analysis for classifier algorithms. Gradient Boosting demonstrated statistically superior performance compared

to other algorithms. AdaBoost, Extra Trees, and Random Forest followed, with no significant differences among them. Multinomial Naive Bayes showed the lowest performance.

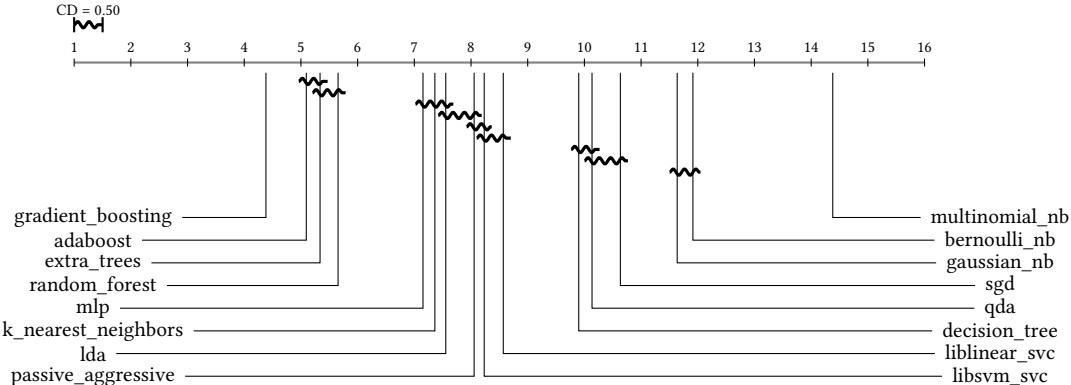

Figure 12: Friedman statistical test and Nemenyi post-hoc analysis for classifiers algorithms ($\alpha = 0.05$). The diagram illustrates statistically significant differences among methods, where groups of methods not connected by a line are significantly different.

Figure 13 presents the analysis of pipeline training time. The fastest algorithms were Gaussian Naive Bayes, followed by Quadratic Discriminant Analysis (QDA). Ensemble methods such as Random Forest, AdaBoost, Extra Trees, and Gradient Boosting had the longest training times. These methods require training multiple base classifiers, unlike single-model classifiers such as Decision Trees. However, when considering the ranking score, Multinomial Naive Bayes emerged as the second fastest.

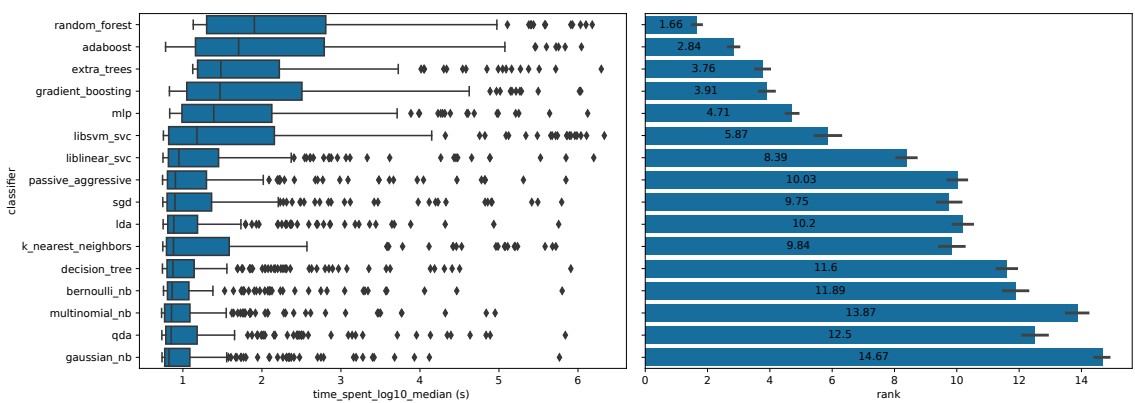

Figure 13: Pipeline training time analysis aggregated by dataset for each classifier using median. The left side displays a boxplot of the time spent, while the right side shows the time-spent rankings. Methods are ordered by the mean time from the boxplot.

Finally, Figure 14 shows the percentage of successful runs, time-out errors, and memory-out errors for each classifier. RF had a 5% memory-out rate, mainly due to its resource-intensive nature when applied to large datasets, as shown in Figure 13. SVM exhibited an 8% memory-out rate, mainly caused by convergence issues that prolonged execution time.

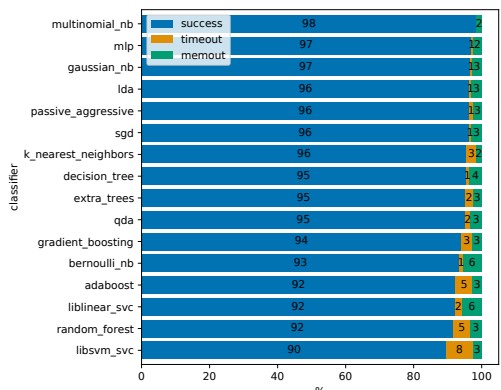

Figure 14: Barplot showing the percentage of successful runs, time-out errors, and memory-out errors for each classifier algorithm.

# C  Configuration Space

Table 1: Configuration space for pipeline experiments (classifiers branch). The first column lists the hyperparameters along with their respective levels (lv) in the tree-based search space. The second column categorizes the hyperparameters. The third column specifies the possible values for each hyperparameter. The final column indicates whether a log10 transformation is applied to the hyperparameter dimension.

| Hyperparameter | | | Type | Dimension | Log Scale |
|---|---|---|---|---|---|
| (lv0) | (lv1) | (lv2) | | | |
| | __choice__ | | categorical | ('adaboost', 'bernoulli_nb', 'decision_tree', 'extra_trees', 'gaussian_nb', 'gradient_boosting', 'k_nearest_neighbors', 'lda', 'liblinear_svc', 'libsvm_svc', 'mlp', 'multinomial_nb', 'passive_aggressive', 'qda', 'random_forest', 'sgd') | - |
| | adaboost | algorithm | categorical | ('SAMME.R', 'SAMME') | - |
| | | learning_rate | uniform float | [0.01, 2.0] | Yes |
| | | max_depth | uniform integer | [1, 10] | No |
| | | n_estimators | uniform integer | [50, 500] | No |
| | bernoulli_nb | alpha | uniform float | [0.01, 100.0] | Yes |
| | | fit_prior | categorical | ('True', 'False') | - |
| | decision_tree | criterion | categorical | ('gini', 'entropy') | - |
| | | max_depth_factor | uniform float | [0.0, 2.0] | No |
| | | max_features | constant | 1 | - |
| | | max_leaf_nodes | constant | None | - |
| | | min_impurity_decrease | constant | 0 | - |
| | | min_samples_leaf | uniform integer | [1, 20] | No |
| | | min_samples_split | uniform integer | [2, 20] | No |
| | | min_weight_fraction_leaf | constant | 0 | - |
| | extra_trees | bootstrap | categorical | ('True', 'False') | - |
| | | criterion | categorical | ('gini', 'entropy') | - |
| | | max_depth | constant | None | - |
| | | max_features | uniform float | [0.0, 1.0] | No |
| | | max_leaf_nodes | constant | None | - |
| | | min_impurity_decrease | constant | 0 | - |
| | | min_samples_leaf | uniform integer | [1, 20] | No |
| | | min_samples_split | uniform integer | [2, 20] | No |
| | | min_weight_fraction_leaf | constant | 0 | - |
| | gradient_boosting | early_stop | categorical | ('off', 'valid', 'train') | - |
| | | l2_regularization | uniform float | [1e-10, 1.0] | Yes |
| classifier | | learning_rate | uniform float | [0.01, 1.0] | Yes |
| | | loss | constant | auto | - |
| | | max_bins | constant | 255 | - |
| | | max_depth | constant | None | - |
| | | max_leaf_nodes | uniform integer | (3, 2047) | Yes |
| | | min_samples_leaf | uniform integer | [1, 200] | Yes |
| | | n_iter_no_change | uniform integer | [1, 20] | No |
| | | scoring | constant | loss | - |
| | | tol | constant | 1.00E-07 | - |
| | | validation_fraction | uniform float | [0.01, 0.4] | No |
| | k_nearest_neighbors | n_neighbors | uniform integer | [1, 100] | Yes |
| | | p | categorical | [1, 2] | - |
| | | weights | categorical | ('uniform', 'distance') | - |
| | lda | shrinkage | categorical | ('None', 'auto', 'manual') | - |
| | | shrinkage_factor | uniform float | [0.0, 1.0] | No |
| | | tol | uniform float | [1e-05, 0.1] | Yes |
| | liblinear_svc | C | uniform float | [0.03125, 32768.0] | Yes |
| | | dual | constant | FALSE | - |
| | | fit_intercept | constant | TRUE | - |
| | | intercept_scaling | constant | 1 | - |
| | | loss | categorical | ('hinge', 'squared_hinge') | - |
| | | multi_class | constant | ovr | - |
| | | penalty | categorical | ('l1', 'l2') | - |
| | | tol | uniform float | [1e-05, 0.1] | Yes |
| | libsvm_svc | C | uniform float | [0.03125, 32768.0] | Yes |
| | | coef0 | uniform float | [-1.0, 1.0] | No |
| | | degree | uniform integer | [2, 5] | No |
| | | gamma | uniform float | [3e-05, 8.0] | Yes |
| | | kernel | categorical | ('rbf', 'poly', 'sigmoid') | - |
| | | max_iter | constant | -1 | - |
| | | shrinking | categorical | ('True', 'False') | - |
| | | tol | uniform float | [1e-05, 0.1] | Yes |
| | mlp | activation | categorical | ('tanh', 'relu') | - |
| | | alpha | uniform float | [1e-07, 0.1] | Yes |

| Hyperparameter | | | Type | Dimension | Log Scale |
|---|---|---|---|---|---|
| (lv0) | (lv1) | (lv2) | | | |
| | | batch_size | constant | auto | - |
| | | beta_1 | constant | 0.9 | - |
| | | beta_2 | constant | 0.999 | - |
| | | early_stopping | categorical | ('valid', 'train') | - |
| | | epsilon | constant | 1.00E-08 | - |
| | | hidden_layer_depth | uniform integer | [1, 3] | No |
| | | learning_rate_init | uniform float | [0.0001, 0.5] | Yes |
| | | n_iter_no_change | constant | 32 | - |
| | | num_nodes_per_layer | uniform integer | [16, 264] | Yes |
| | | shuffle | constant | TRUE | - |
| | | solver | constant | adam | - |
| | | tol | constant | 0.0001 | - |
| | | validation_fraction | constant | 0.1 | - |
| | multinomial_nb | alpha | uniform float | [0.01, 100.0] | Yes |
| | | fit_prior | categorical | ('True', 'False') | - |
| | passive_aggressive | C | uniform float | [1e-05, 10.0] | Yes |
| | | average | categorical | ('False', 'True') | - |
| | | fit_intercept | constant | TRUE | - |
| | | loss | categorical | ('hinge', 'squared_hinge') | - |
| | | tol | uniform float | [1e-05, 0.1] | Yes |
| | qda | reg_param | uniform float | [0.0, 1.0] | No |
| | random_forest | bootstrap | categorical | ('True', 'False') | - |
| | | criterion | categorical | ('gini', 'entropy') | - |
| | | max_depth | constant | None | - |
| | | max_features | uniform float | [0.0, 1.0] | No |
| | | max_leaf_nodes | constant | None | - |
| | | min_impurity_decrease | constant | 0 | - |
| | | min_samples_leaf | uniform integer | [1, 20] | No |
| | | min_samples_split | uniform integer | [2, 20] | No |
| | | min_weight_fraction_leaf | constant | 0 | - |
| | sgd | alpha | uniform float | [1e-07, 0.1] | Yes |
| | | average | categorical | ('False', 'True') | - |
| | | epsilon | uniform float | [1e-05, 0.1] | Yes |
| | | eta0 | uniform float | [1e-07, 0.1] | Yes |
| | | fit_intercept | constant | TRUE | - |
| | | l1_ratio | uniform float | [1e-09, 1.0] | Yes |
| | | learning_rate | categorical | ('optimal', 'invscaling', 'constant') | - |
| | | loss | categorical | ('hinge', 'log', 'modified_huber', 'squared_hinge', 'perceptron') | - |
| | | penalty | categorical | ('l1', 'l2', 'elasticnet') | - |
| | | power_t | uniform float | [1e-05, 1.0] | No |
| | | tol | uniform float | [1e-05, 0.1] | Yes |
| balancing | strategy | | categorical | ('none', 'weighting') | - |

Table 2: Configuration space for pipeline experiments (feature preprocessing branch). The first column lists the hyperparameters along with their respective levels (lv) in the tree-based search space. The second column categorizes the hyperparameters. The third column specifies the possible values for each hyperparameter. The final column indicates whether a log10 transformation is applied to the hyperparameter dimension.

| Hyperparameter | | | Type | Dimension | Log Scale |
|---|---|---|---|---|---|
| (lv0) | (lv1) | (lv2) | | | |
| | __choice__ | | categorical | ('extra_trees_preproc', 'fast_ica', 'feature_agglomeration', 'kernel_pca', 'kitchen_sinks', 'liblinear_svc_preprocessor', 'no_preprocessing', 'nystroem_sampler', 'pca', 'polynomial', 'random_trees_embedding', 'select_percentile', 'select_rates') | - |
| | extra_trees_preproc | bootstrap | categorical | ('True', 'False') | - |
| | | criterion | categorical | ('gini', 'entropy') | - |
| | | max_depth | constant | None | - |
| | | max_features | uniform float | [0.0, 1.0] | No |
| | | max_leaf_nodes | constant | None | - |
| | | min_impurity_decrease | constant | 0 | - |
| | | min_samples_leaf | uniform integer | [1, 20] | No |
| | | min_samples_split | uniform integer | [2, 20] | No |

| Hyperparameter | | | Type | Dimension | Log Scale |
|---|---|---|---|---|---|
| (lv0) | (lv1) | (lv2) | | | |
| | | min_weight_fraction_leaf | constant | 0 | - |
| | | n_estimators | constant | 100 | - |
| | fast_ica | algorithm | categorical | ('parallel', 'deflation') | - |
| | | fun | categorical | ('logcosh', 'exp', 'cube') | - |
| | | n_components | uniform integer | [10, 2000] | No |
| | | whiten | categorical | ('False', 'True') | - |
| | feature_agglomeration | affinity | categorical | ('euclidean', 'manhattan', 'cosine') | - |
| | | linkage | categorical | ('ward', 'complete', 'average') | - |
| | | n_clusters | uniform integer | [2, 400] | No |
| | | pooling_func | categorical | ('mean', 'median', 'max') | - |
| | kernel_pca | coef0 | uniform float | [-1.0, 1.0] | No |
| | | degree | uniform integer | [2, 5] | No |
| | | gamma | uniform float | [3e-05, 8.0] | Yes |
| | | kernel | categorical | ('poly', 'rbf', 'sigmoid', 'cosine') | - |
| | | n_components | uniform integer | [10, 2000] | No |
| | kitchen_sinks | gamma | uniform float | [3e-05, 8.0] | Yes |
| | | n_components | uniform integer | [50, 10000] | Yes |
| | liblinear_svc_preprocessor | C | uniform float | [0.03125, 32768.0] | Yes |
| | | dual | constant | FALSE | - |
| | | fit_intercept | constant | TRUE | - |
| | | intercept_scaling | constant | 1 | - |
| | | loss | categorical | ('hinge', 'squared_hinge') | - |
| | | multi_class | constant | ovr | - |
| | | penalty | constant | l1 | - |
| | | tol | uniform float | [1e-05, 0.1] | Yes |
| | nystroem_sampler | coef0 | uniform float | [-1.0, 1.0] | No |
| | | degree | uniform integer | [2, 5] | No |
| | | gamma | uniform float | [3e-05, 8.0] | Yes |
| | | kernel | categorical | ('poly', 'rbf', 'sigmoid', 'cosine') | - |
| | | n_components | uniform integer | [50, 10000] | Yes |
| | pca | keep_variance | uniform float | [0.5, 0.9999] | No |
| | | whiten | categorical | ('False', 'True') | - |
| | polynomial | degree | uniform integer | [2, 3] | No |
| | | include_bias | categorical | ('True', 'False') | - |
| | | interaction_only | categorical | ('False', 'True') | - |
| | random_trees_embedding | bootstrap | categorical | ('True', 'False') | - |
| | | max_depth | uniform integer | [2, 10] | No |
| | | max_leaf_nodes | constant | None | - |
| | | min_samples_leaf | uniform integer | [1, 20] | No |
| | | min_samples_split | uniform integer | [2, 20] | No |
| | | min_weight_fraction_leaf | constant | 1 | - |
| | | n_estimators | uniform integer | [10, 100] | No |
| | select_percentile | percentile | uniform float | [1.0, 99.0] | No |
| | | score_func | categorical | ('chi2', 'f_classif', 'mutual_info') | - |
| | select_rates | alpha | uniform float | [0.01, 0.5] | No |
| | | mode | categorical | ('fpr', 'fdr', 'fwe') | - |
| | | score_func | categorical | ('chi2', 'f_classif', 'mutual_info_classif') | - |

Table 3: Configuration space for pipeline experiments (data preprocessing branch). The first column lists the hyperparameters along with their respective levels (lv) in the tree-based search space. The second column categorizes the hyperparameters. The third column specifies the possible values for each hyperparameter. The final column indicates whether a log10 transformation is applied to the hyperparameter dimension. The categorical transform is used over while the numerical transform is the opposite. Categorical transformation and imputation are always required due to sklearn limitations.

| Hyperparameter (lv0) | (lv1) | (lv2) | (lv3) | (lv4) | (lv5) | Type | Dimension | Log Scale |
|---|---|---|---|---|---|---|---|---|
| data_preprocessor | __choice__ | | | | | categorical | ('feature_type',) | - |
| | | categorical_transformer | categorical_encoding | __choice__ | | categorical | ('encoding', 'no_encoding', 'one_hot_encoding') | - |
| feature_type | | | category_coalescence | __choice__ | | categorical | ('minority_coalescer', 'no_coalescense') | - |
| | | | category_coalescence | minority_coalescer | minimum_fraction | uniform float | [0.0001, 0.5] | Yes |
| | | | imputation | strategy | | categorical | ('mean', 'median', 'most_frequent') | - |
| | | numerical_transformer | | __choice__ | | categorical | ('minmax', 'none', 'normalize', 'power_transformer', 'quantile_transformer', 'robust_scaler', 'standardize') | - |
| | | | rescaling | quantile_transformer | n_quantiles | uniform integer | [10, 2000] | No |
| | | | | quantile_transformer | output_distribution | categorical | ('normal', 'uniform') | - |
| | | | | robust_scaler | q_max | uniform float | [0.7, 0.999] | No |
| | | | | robust_scaler | q_min | uniform float | [0.001, 0.3] | No |

# D  Performance Aggregated by Mean and Median

In this section, we presented the performance aggregated by mean and median for feature preprocessing, classifier and pipeline.

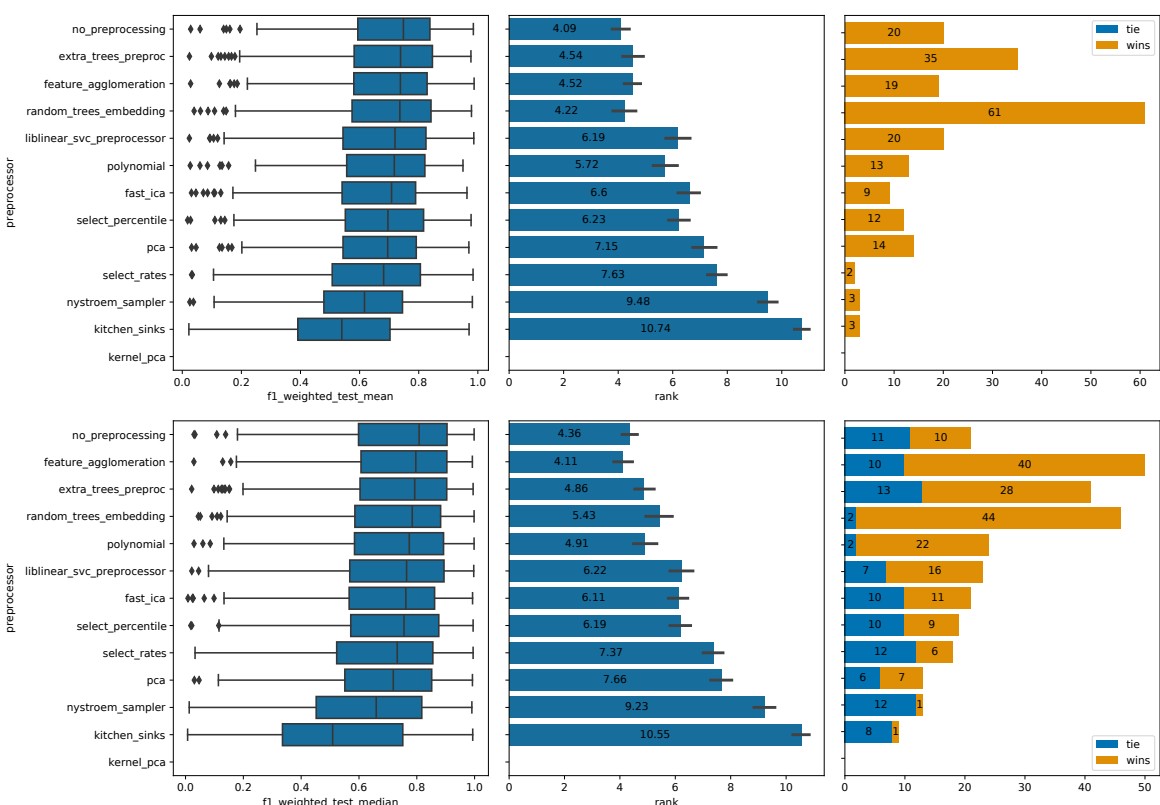

Figure 15: Aggregated F1 by dataset for each for each feature preprocessor. From top to bottom, the figure shows F1 performance aggregated using mean and median. From left to right, the plots display the performance boxplots, the ranking barplot, and the wins and ties barplot for each method. All plots are ordered by boxplot mean.

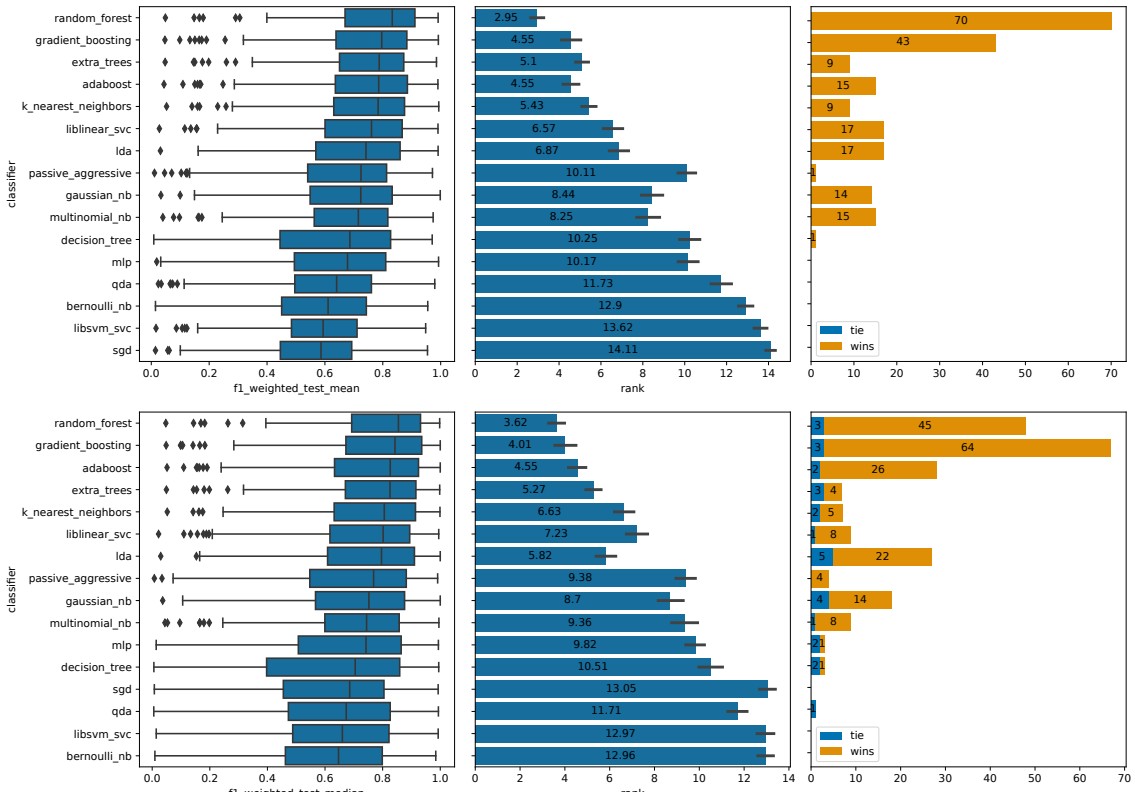

Figure 16: Aggregated F1 by dataset for each for each classifier algorithm. From top to bottom, the figure shows F1 performance aggregated using mean and median. From left to right, the plots display the performance boxplots, the ranking barplot, and the wins and ties barplot for each classifier. All plots are ordered by boxplot mean.

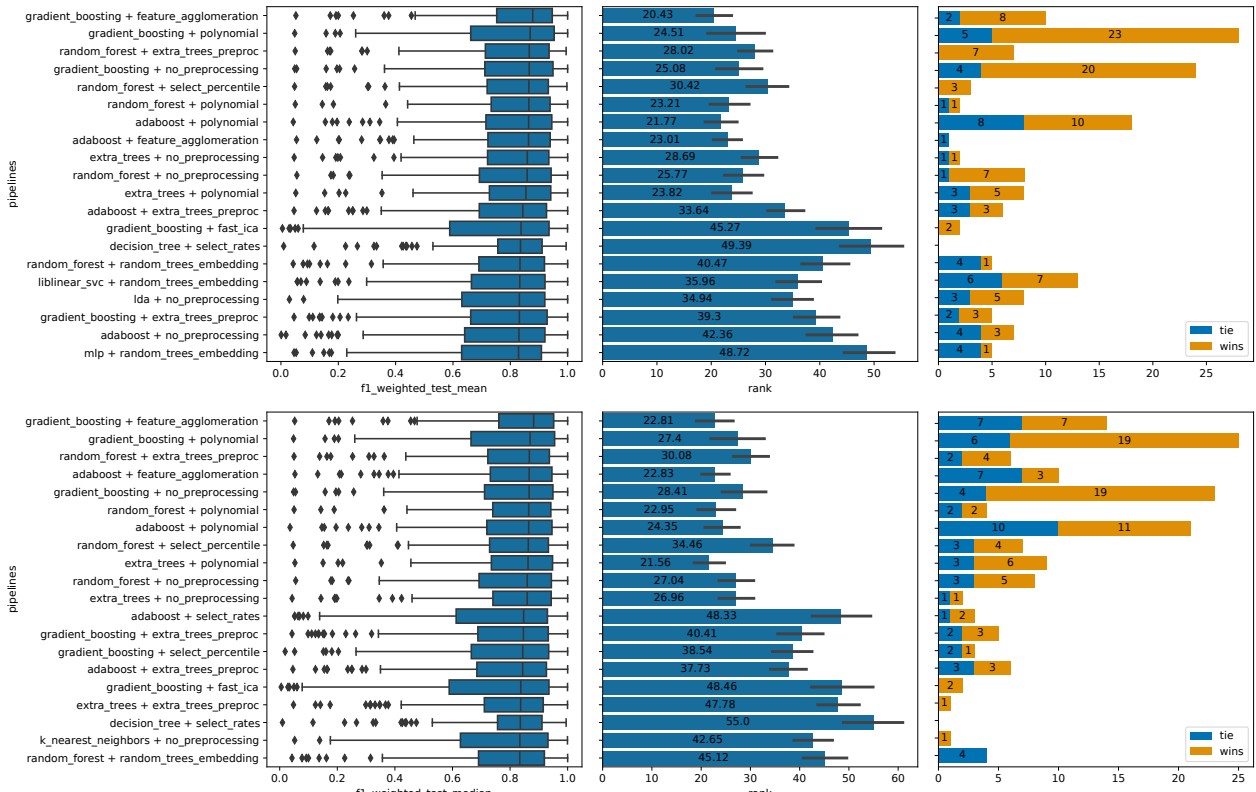

Figure 17: Aggregated F1 by dataset for each for each machine learning pipeline. From top to bottom, the figure shows F1 performance aggregated using mean and median. From left to right, the plots display the performance boxplots, the ranking barplot, and the wins and ties barplot for each pipeline. All plots are ordered by boxplot mean. Only the top 20 outperforming, sorted by boxplot mean, were selected.

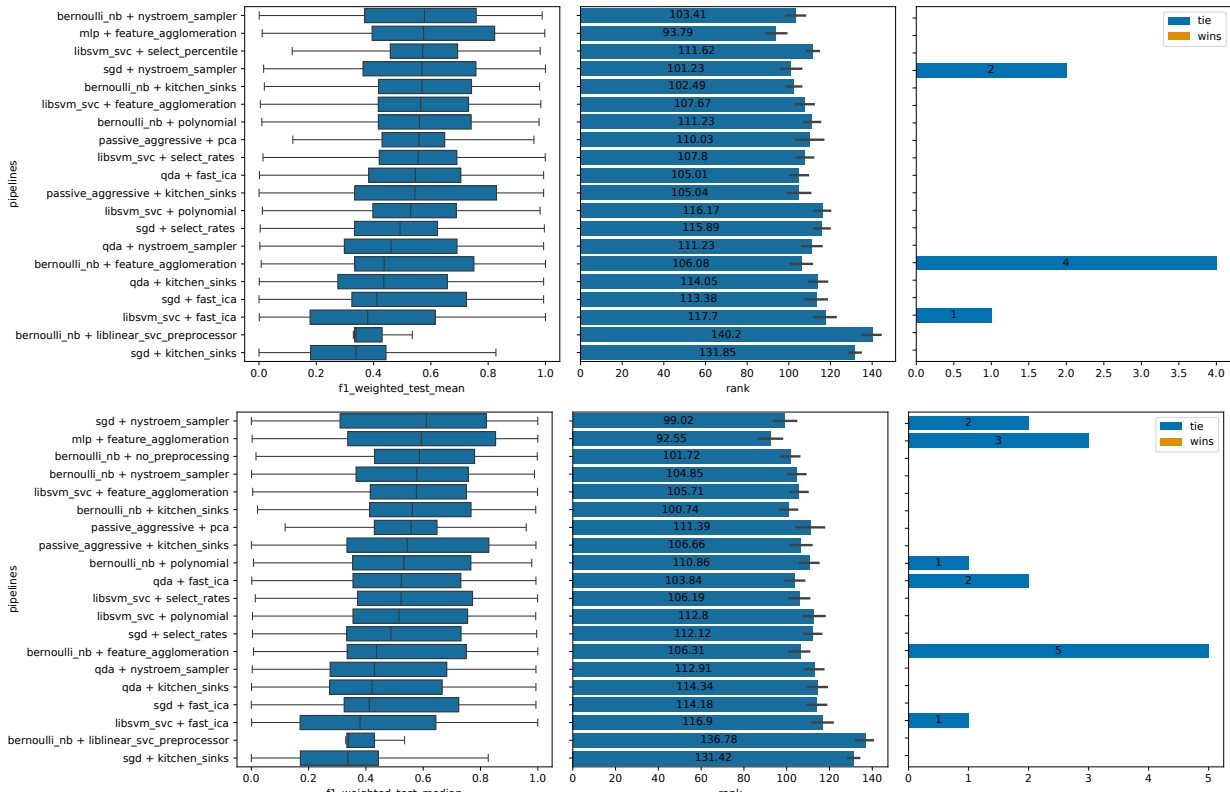

Figure 18: Aggregated F1 by dataset for each for each machine learning pipeline. From top to bottom, the figure shows F1 performance aggregated using mean and median deviation. From left to right, the plots display the performance boxplots, the ranking barplot, and the wins and ties barplot for each pipeline. All plots are ordered by boxplot mean. Only the top 20 worst minus Kernel PCA, sorted by boxplot mean, were selected.

# E Datasets

Table 4: Datasets used in the study. The columns, from left to right, show: (i) the OpenML dataset ID, (ii) the number of examples, (iii) the number of features, (iv) the number of categorical features, (v) the number of classes, (vi) the percentage of examples in the majority class, and (vii) the percentage of examples in the minority class. Percentage less than 0.01% is indicate with *.

| OpenML ID | # Examples | # Features | # Categorical Features | Number of class | Majority Class % | Minority Class % |
|---|---|---|---|---|---|---|
| 2 | 898 | 38 | 32 | 5 | 76.17 | 0.00* |
| 6 | 20000 | 16 | 0 | 26 | 4.070 | 3.67 |
| 11 | 625 | 4 | 0 | 3 | 46.08 | 7.84 |
| 15 | 699 | 9 | 0 | 2 | 65.52 | 34.48 |
| 23 | 1473 | 9 | 7 | 3 | 42.70 | 22.61 |
| 24 | 8124 | 22 | 22 | 2 | 51.80 | 48.20 |
| 26 | 12960 | 8 | 8 | 5 | 33.33 | 0.02 |
| 28 | 5620 | 64 | 0 | 10 | 10.18 | 9.86 |
| 30 | 5473 | 10 | 0 | 5 | 89.77 | 0.51 |
| 32 | 10992 | 16 | 0 | 10 | 10.41 | 9.60 |
| 37 | 768 | 8 | 0 | 2 | 65.10 | 34.90 |
| 42 | 683 | 35 | 35 | 19 | 13.47 | 1.17 |
| 44 | 4601 | 57 | 0 | 2 | 60.60 | 39.40 |
| 46 | 3190 | 60 | 60 | 3 | 51.88 | 24.04 |
| 50 | 958 | 9 | 9 | 2 | 65.34 | 34.66 |
| 57 | 3772 | 29 | 22 | 4 | 92.29 | 0.05 |
| 60 | 5000 | 40 | 0 | 3 | 33.84 | 33.06 |
| 151 | 45312 | 8 | 1 | 2 | 57.55 | 42.45 |
| 155 | 829201 | 10 | 5 | 10 | 50.11 | 0.00* |
| 181 | 1484 | 8 | 0 | 10 | 31.20 | 0.34 |
| 182 | 6430 | 36 | 0 | 6 | 23.81 | 9.72 |
| 184 | 28056 | 6 | 6 | 18 | 16.23 | 0.10 |
| 185 | 1340 | 16 | 1 | 3 | 90.67 | 4.25 |
| 188 | 736 | 19 | 5 | 5 | 29.08 | 14.27 |
| 279 | 45164 | 74 | 0 | 11 | 50.97 | 0.00* |
| 300 | 7797 | 617 | 0 | 26 | 3.85 | 3.82 |
| 307 | 990 | 12 | 2 | 11 | 9.09 | 9.09 |
| 310 | 11183 | 6 | 0 | 2 | 97.68 | 2.32 |
| 311 | 937 | 49 | 0 | 2 | 95.62 | 4.38 |
| 333 | 556 | 6 | 6 | 2 | 50.00 | 50.00 |
| 334 | 601 | 6 | 6 | 2 | 65.72 | 34.28 |
| 335 | 554 | 6 | 6 | 2 | 51.99 | 48.01 |
| 375 | 9961 | 14 | 0 | 9 | 16.20 | 7.85 |
| 377 | 600 | 60 | 0 | 6 | 16.67 | 16.67 |
| 451 | 500 | 5 | 3 | 2 | 55.60 | 44.40 |
| 458 | 841 | 70 | 0 | 4 | 37.69 | 6.54 |
| 469 | 797 | 4 | 4 | 6 | 19.45 | 15.43 |
| 470 | 672 | 9 | 4 | 2 | 66.67 | 33.33 |
| 554 | 70000 | 784 | 0 | 10 | 11.25 | 9.02 |
| 715 | 1000 | 25 | 0 | 2 | 55.70 | 44.30 |
| 717 | 508 | 10 | 0 | 2 | 56.30 | 43.70 |
| 722 | 15000 | 48 | 0 | 2 | 66.39 | 33.61 |
| 725 | 8192 | 8 | 0 | 2 | 59.63 | 40.37 |
| 727 | 40768 | 10 | 0 | 2 | 50.09 | 49.91 |
| 728 | 4052 | 7 | 0 | 2 | 76.04 | 23.96 |
| 734 | 13750 | 40 | 0 | 2 | 57.61 | 42.39 |
| 735 | 8192 | 12 | 0 | 2 | 69.76 | 30.24 |
| 737 | 3107 | 6 | 0 | 2 | 50.40 | 49.60 |

| OpenML ID | # Examples | # Features | # Categorical Features | Number of class | Majority Class % | Minority Class % |
|---|---|---|---|---|---|---|
| 740 | 1000 | 10 | 0 | 2 | 56.00 | 44.00 |
| 742 | 500 | 100 | 0 | 2 | 56.60 | 43.40 |
| 750 | 500 | 7 | 0 | 2 | 50.80 | 49.20 |
| 752 | 8192 | 32 | 0 | 2 | 50.39 | 49.61 |
| 757 | 528 | 21 | 2 | 2 | 89.77 | 10.23 |
| 761 | 8192 | 21 | 0 | 2 | 69.76 | 30.24 |
| 770 | 625 | 6 | 0 | 2 | 50.40 | 49.60 |
| 772 | 2178 | 3 | 0 | 2 | 55.51 | 44.49 |
| 799 | 1000 | 5 | 0 | 2 | 50.30 | 49.70 |
| 802 | 1945 | 18 | 6 | 2 | 50.03 | 49.97 |
| 803 | 7129 | 5 | 0 | 2 | 53.06 | 46.94 |
| 807 | 8192 | 8 | 0 | 2 | 50.88 | 49.12 |
| 816 | 8192 | 8 | 0 | 2 | 50.22 | 49.78 |
| 819 | 9517 | 6 | 0 | 2 | 50.28 | 49.72 |
| 821 | 22784 | 16 | 0 | 2 | 70.40 | 29.60 |
| 823 | 20640 | 8 | 0 | 2 | 56.81 | 43.19 |
| 825 | 506 | 20 | 3 | 2 | 55.93 | 44.07 |
| 826 | 576 | 11 | 11 | 2 | 58.51 | 41.49 |
| 833 | 8192 | 32 | 0 | 2 | 68.96 | 31.04 |
| 837 | 1000 | 50 | 0 | 2 | 54.70 | 45.30 |
| 839 | 782 | 8 | 2 | 2 | 64.96 | 35.04 |
| 841 | 950 | 9 | 0 | 2 | 51.37 | 48.63 |
| 846 | 16599 | 18 | 0 | 2 | 69.09 | 30.91 |
| 847 | 6574 | 14 | 0 | 2 | 53.26 | 46.74 |
| 871 | 3848 | 5 | 0 | 2 | 50.00 | 50.00 |
| 881 | 40768 | 10 | 3 | 2 | 59.66 | 40.34 |
| 884 | 500 | 5 | 0 | 2 | 50.20 | 49.80 |
| 886 | 500 | 7 | 0 | 2 | 50.20 | 49.80 |
| 897 | 1161 | 15 | 2 | 2 | 70.03 | 29.97 |
| 901 | 40768 | 10 | 0 | 2 | 50.11 | 49.89 |
| 903 | 1000 | 25 | 0 | 2 | 56.30 | 43.70 |
| 920 | 500 | 50 | 0 | 2 | 59.00 | 41.00 |
| 923 | 8641 | 4 | 1 | 2 | 55.01 | 44.99 |
| 930 | 1302 | 33 | 1 | 2 | 52.84 | 47.16 |
| 934 | 1156 | 5 | 4 | 2 | 77.85 | 22.15 |
| 936 | 500 | 10 | 0 | 2 | 54.40 | 45.60 |
| 937 | 500 | 50 | 0 | 2 | 56.40 | 43.60 |
| 940 | 527 | 36 | 15 | 2 | 84.82 | 15.18 |
| 947 | 559 | 4 | 1 | 2 | 95.71 | 4.29 |
| 949 | 559 | 4 | 1 | 2 | 85.69 | 14.31 |
| 950 | 559 | 4 | 1 | 2 | 96.60 | 3.40 |
| 951 | 559 | 4 | 1 | 2 | 97.67 | 2.33 |
| 981 | 10108 | 68 | 68 | 2 | 73.14 | 26.86 |
| 1039 | 4229 | 1617 | 0 | 2 | 96.48 | 3.52 |
| 1044 | 10936 | 27 | 3 | 3 | 38.97 | 26.24 |
| 1046 | 15545 | 5 | 0 | 2 | 67.14 | 32.86 |
| 1049 | 1458 | 37 | 0 | 2 | 87.79 | 12.21 |
| 1050 | 1563 | 37 | 0 | 2 | 89.76 | 10.24 |
| 1053 | 10885 | 21 | 0 | 2 | 80.65 | 19.35 |
| 1056 | 9466 | 38 | 0 | 2 | 99.28 | 0.72 |
| 1063 | 522 | 21 | 0 | 2 | 79.50 | 20.50 |
| 1068 | 1109 | 21 | 0 | 2 | 93.06 | 6.94 |
| 1069 | 5589 | 36 | 0 | 2 | 99.59 | 0.41 |
| 1116 | 6598 | 167 | 1 | 2 | 84.59 | 15.41 |
| 1120 | 19020 | 10 | 0 | 2 | 64.84 | 35.16 |
| 1128 | 1545 | 10935 | 0 | 2 | 77.73 | 22.27 |

| OpenML ID | # Examples | # Features | # Categorical Features | Number of class | Majority Class % | Minority Class % |
|---|---|---|---|---|---|---|
| 1130 | 1545 | 10935 | 0 | 2 | 91.84 | 8.16 |
| 1134 | 1545 | 10935 | 0 | 2 | 83.17 | 16.83 |
| 1142 | 1545 | 10935 | 0 | 2 | 96.05 | 3.95 |
| 1146 | 1545 | 10935 | 0 | 2 | 95.53 | 4.47 |
| 1161 | 1545 | 10935 | 0 | 2 | 81.49 | 18.51 |
| 1166 | 1545 | 10935 | 0 | 2 | 87.18 | 12.82 |
| 1233 | 945 | 6373 | 0 | 7 | 14.81 | 12.59 |
| 1457 | 1500 | 10000 | 0 | 50 | 2.00 | 2.00 |
| 1459 | 10218 | 7 | 0 | 10 | 13.86 | 5.87 |
| 1462 | 1372 | 4 | 0 | 2 | 55.54 | 44.46 |
| 1466 | 2126 | 35 | 0 | 10 | 27.23 | 2.49 |
| 1471 | 14980 | 14 | 0 | 2 | 55.12 | 44.88 |
| 1475 | 6118 | 51 | 0 | 6 | 41.75 | 7.94 |
| 1478 | 10299 | 561 | 0 | 6 | 18.88 | 13.65 |
| 1479 | 1212 | 100 | 0 | 2 | 50.00 | 50.00 |
| 1480 | 583 | 10 | 1 | 2 | 71.36 | 28.64 |
| 1481 | 28056 | 6 | 3 | 18 | 16.23 | 0.10 |
| 1483 | 164860 | 7 | 2 | 11 | 33.05 | 0.84 |
| 1485 | 2600 | 500 | 0 | 2 | 50.00 | 50.00 |
| 1487 | 2534 | 72 | 0 | 2 | 93.69 | 6.31 |
| 1491 | 1600 | 64 | 0 | 100 | 1.00 | 1.00 |
| 1494 | 1055 | 41 | 0 | 2 | 66.26 | 33.74 |
| 1496 | 7400 | 20 | 0 | 2 | 50.49 | 49.51 |
| 1497 | 5456 | 24 | 0 | 4 | 40.41 | 6.01 |
| 1501 | 1593 | 256 | 0 | 10 | 10.17 | 9.73 |
| 1502 | 245057 | 3 | 0 | 2 | 79.25 | 20.75 |
| 1503 | 263256 | 14 | 0 | 10 | 10.06 | 9.92 |
| 1507 | 7400 | 20 | 0 | 2 | 50.04 | 49.96 |
| 1509 | 149332 | 4 | 0 | 22 | 14.73 | 0.61 |
| 1510 | 569 | 30 | 0 | 2 | 62.74 | 37.26 |
| 1515 | 571 | 1300 | 0 | 20 | 10.51 | 1.93 |
| 1528 | 1623 | 3 | 0 | 5 | 90.63 | 1.79 |
| 1529 | 1521 | 3 | 0 | 5 | 90.01 | 1.91 |
| 1530 | 1515 | 3 | 0 | 5 | 90.10 | 1.91 |
| 1531 | 10176 | 3 | 0 | 5 | 96.22 | 0.26 |
| 1532 | 10668 | 3 | 0 | 5 | 96.41 | 0.24 |
| 1535 | 9989 | 3 | 0 | 5 | 96.10 | 0.26 |
| 1536 | 10130 | 3 | 0 | 5 | 96.21 | 0.26 |
| 1538 | 8753 | 3 | 0 | 5 | 94.42 | 0.64 |
| 1541 | 8654 | 3 | 0 | 5 | 94.33 | 0.65 |
| 1542 | 1183 | 3 | 0 | 5 | 91.55 | 0.76 |
| 1547 | 1000 | 20 | 0 | 2 | 74.10 | 25.90 |
| 1549 | 750 | 40 | 3 | 8 | 22.00 | 7.60 |
| 1552 | 1100 | 12 | 4 | 5 | 27.73 | 13.91 |
| 1553 | 700 | 12 | 4 | 3 | 35.00 | 30.57 |
| 1590 | 48842 | 14 | 8 | 2 | 76.07 | 23.93 |
| 4134 | 3751 | 1776 | 0 | 2 | 54.23 | 45.77 |
| 4534 | 11055 | 30 | 30 | 2 | 55.69 | 44.31 |
| 4538 | 9873 | 32 | 0 | 5 | 29.88 | 10.11 |
| 4541 | 101766 | 49 | 36 | 3 | 53.91 | 11.16 |
| 6332 | 540 | 37 | 19 | 2 | 57.78 | 42.22 |
| 23380 | 2796 | 33 | 2 | 6 | 24.32 | 9.80 |
| 23381 | 500 | 12 | 11 | 2 | 58.00 | 42.00 |
| 40496 | 500 | 7 | 0 | 10 | 11.40 | 7.40 |
| 40498 | 4898 | 11 | 0 | 7 | 44.88 | 0.10 |
| 40499 | 5500 | 40 | 0 | 11 | 9.09 | 9.09 |

| OpenML ID | # Examples | # Features | # Categorical Features | Number of class | Majority Class % | Minority Class % |
|---|---|---|---|---|---|---|
| 40536 | 8378 | 120 | 61 | 2 | 83.53 | 16.47 |
| 40645 | 1600 | 1000 | 1000 | 2 | 50.00 | 50.00 |
| 40646 | 1600 | 20 | 20 | 2 | 50.00 | 50.00 |
| 40647 | 1600 | 20 | 20 | 2 | 50.00 | 50.00 |
| 40648 | 1600 | 20 | 20 | 2 | 50.00 | 50.00 |
| 40649 | 1600 | 20 | 20 | 2 | 50.00 | 50.00 |
| 40650 | 1600 | 20 | 20 | 2 | 50.00 | 50.00 |
| 40668 | 67557 | 42 | 42 | 3 | 65.83 | 9.55 |
| 40670 | 3186 | 180 | 180 | 3 | 51.91 | 24.01 |
| 40672 | 100968 | 29 | 15 | 8 | 41.71 | 0.010 |
| 40677 | 3200 | 24 | 24 | 10 | 10.53 | 9.25 |
| 40680 | 1324 | 10 | 10 | 2 | 77.95 | 22.05 |
| 40691 | 1599 | 11 | 0 | 6 | 42.59 | 0.63 |
| 40693 | 973 | 9 | 9 | 2 | 66.91 | 33.09 |
| 40701 | 5000 | 20 | 4 | 2 | 85.86 | 14.14 |
| 40704 | 2201 | 3 | 0 | 2 | 67.70 | 32.30 |
| 40705 | 959 | 44 | 2 | 2 | 63.92 | 36.08 |
| 40706 | 1124 | 10 | 10 | 2 | 50.44 | 49.56 |
| 40900 | 5100 | 36 | 0 | 2 | 98.53 | 1.47 |
| 40922 | 88588 | 6 | 0 | 2 | 50.08 | 49.92 |
| 40923 | 92000 | 1024 | 0 | 46 | 2.17 | 2.17 |
| 40927 | 60000 | 3072 | 0 | 10 | 10.00 | 10.00 |
| 40966 | 1080 | 77 | 0 | 8 | 13.89 | 9.72 |
| 40971 | 1000 | 19 | 0 | 30 | 8.00 | 0.60 |
| 40978 | 3279 | 1558 | 1555 | 2 | 86.00 | 14.00 |
| 40982 | 1941 | 27 | 0 | 7 | 34.67 | 2.83 |
| 40983 | 4839 | 5 | 0 | 2 | 94.61 | 5.39 |
| 40985 | 45781 | 2 | 0 | 20 | 6.35 | 3.05 |
| 40994 | 540 | 18 | 0 | 2 | 91.48 | 8.52 |
| 41082 | 9298 | 256 | 0 | 10 | 16.7 | 7.61 |
| 41084 | 575 | 10304 | 0 | 20 | 8.35 | 3.3 |
| 41144 | 3140 | 259 | 0 | 2 | 50.29 | 49.71 |
| 41145 | 5832 | 308 | 0 | 2 | 50.00 | 50.00 |
| 41146 | 5124 | 20 | 0 | 2 | 50.00 | 50.00 |
| 41147 | 425240 | 78 | 52 | 2 | 50.00 | 50.00 |
| 41150 | 130064 | 50 | 0 | 2 | 71.94 | 28.06 |
| 41160 | 31406 | 22 | 14 | 2 | 90.46 | 9.54 |
| 41162 | 72983 | 32 | 18 | 2 | 87.70 | 12.30 |
| 41163 | 10000 | 2000 | 0 | 5 | 20.49 | 19.13 |
| 41671 | 20000 | 20 | 0 | 5 | 55.81 | 3.72 |
| 41972 | 9144 | 220 | 0 | 8 | 44.29 | 0.22 |
| 41982 | 70000 | 784 | 0 | 10 | 10.00 | 10.00 |
| 41986 | 51839 | 1568 | 0 | 43 | 5.79 | 0.52 |
| 41988 | 51839 | 1568 | 0 | 43 | 5.79 | 0.52 |
| 41989 | 51839 | 2916 | 0 | 43 | 5.79 | 0.52 |
| 41990 | 51839 | 256 | 0 | 43 | 5.79 | 0.52 |
| 41991 | 270912 | 784 | 0 | 49 | 2.58 | 0.17 |
| 42193 | 5278 | 13 | 6 | 2 | 52.96 | 47.04 |
| 42206 | 595212 | 37 | 25 | 2 | 96.36 | 3.64 |
| 42343 | 82318 | 477 | 136 | 2 | 88.23 | 11.77 |
| 42345 | 70340 | 20 | 19 | 3 | 48.88 | 4.98 |

## F  Performance Tables

The performance tables that generated the figures are available on GitHub in the path "*analysis/base-level-analysis/performance_avaluation*"[3]. You can also find different performance metrics on this path as well.

---

[3]https://github.com/ealcobaca/exploring-machine-learning-pipelines/tree/main/analysis/base-level-analysis/perforamnce_evaluation

