# OpenReview forum: "Exploring One Million Machine Learning Pipelines: A Benchmarking Study"
_automl.cc/AutoML/2025/ABCD_Track — AutoML 2025 ABCD Track_

### Official Review · Reviewer_ZuFa · 2025-04-10

**Comments To Authors:**

This manuscript presents an ambitious and valuable benchmarking effort, systematically evaluating over one million machine learning pipelines formed by varying combinations of preprocessing techniques and classification algorithms across 211 diverse datasets. By adopting a random search strategy, the authors mitigate selection bias typically introduced by model-based optimization, thereby enabling an assumption-free exploration of the configuration space. The resulting meta-knowledge dataset is of substantial relevance to the AutoML and meta-learning communities, offering a reproducible and extensible foundation for future research on pipeline selection and automated configuration.

The contribution is timely and significant, particularly in its scale and its effort to reduce optimization-induced artifacts. The generation of 1,055,000 pipeline evaluations and their systematic aggregation represents a strong empirical foundation. Additionally, the commitment to open data and code promotes transparency and reproducibility, which are critical for the progress of AutoML research.

However, several limitations merit consideration:

- While random search is appropriate for unbiased sampling, the absence of fine-grained hyperparameter optimization likely penalizes tuning-sensitive models such as SVMs, MLPs, and kernel-based classifiers. Their underperformance in the benchmark may reflect configuration artifacts rather than intrinsic limitations, thus skewing comparative conclusions.

- The reliance on Auto-sklearn’s internal configuration limits the scope of the study. Notably, it excludes recent model classes (e.g., transformer-based tabular models, deep ensembles) and more sophisticated preprocessing operators that are now emerging in contemporary pipelines.

- Although the paper aims to support meta-learning, it stops short of demonstrating how the provided dataset improves downstream meta-model performance. An illustrative use case or baseline meta-learner evaluation would significantly enhance the practical impact of the contribution.

- Preprocessing components are sampled agnostically, yet some preprocessing-classifier interactions have well-known dependencies (e.g., standardization before SVM, one-hot encoding incompatibility with certain tree-based methods). The lack of semantic constraints in the pipeline generation process risks producing implausible or functionally suboptimal configurations.

These concerns, while not diminishing the value of the work, should be acknowledged more explicitly. Addressing them—either through additional experiments or extended discussion—would further strengthen the paper’s robustness and broaden its applicability.

**Review Confidence:**

5

**Review Rating:**

7

---

### Official Review · Reviewer_yVMy · 2025-04-27

**Comments To Authors:**

## Summary
This paper examines the impact of different setups of machine learning pipeline configurations, each consisting of data preprocessing, feature preprocessing and a classifier. The study is evaluated across a wide range of 211 tabular classification datasets, finding that ensemble tree-based models - particularly Extra Trees, Random Forest, AdaBoost and Gradient Boosting consistently outperform other methods, often even without preprocessing. In contrast, kernel-based methods  and Naive Bayes tend to underperform, likely due to sensitivity to hyperparamter configurations and limited model assumptions. The results offer practical guidance for designing AutoML pipelines, providing both practitioners and researchers with a comprehensive benchmark of model and preprocessing combinations.

## Strengths
- An extensive benchmark on 211 tabular classification datasets of different domains, which strengthens the generalizability and reliability of the findings. Tested on a wide range of preprocessing and modeling techniques, many of which heavily used for tabular data problems.
- Providing a good overview of practical design principles for AutoML systems, especially by presenting a comparison of different modeling and preprocessing techniques indicating when (and if so, which) preprocessing might be more or less beneficial.
- Consideration of predictive performance metrics (F1-score) for different pipelines but also efficiency in the form of average training time per pipeline. Additionally the amount of times where training of a pipeline fails due to memory or time-out errors is considered.

## Weaknesses
- The comparison of different pipelines mainly shows ensemble tree-based methods to outperform other methods, regardless of the type of preprocessing applied in many cases. While the benchmark of such methods in combination with different preprocessing techniques is a valuable contribution as mentioned in the strengths section, it largely re-affirms established understanding. [1],[2]
- Some findings mentioned in Section 4.2 are redundant or self-evident, such as the ineffectiveness of a lot of feature preprocessing techniques for tree-based methods, which are well known to internally perform perform feature selection or reduction.
- In Section 3.3 it states 500 x 211 x 10 pipeline evaluations leading to a total of 1.055.000 pipeline evaluations however the 10 in the calculation is seemingly from the CV folds on the training set, which only leads to 500 x 211 individual pipelines (pipelines of different configurations)
- The used models for the evaluation represent a limited subset of existing and state-of-the-art models. Especially more powerful models such as XGBoost, LightGBM or CatBoost would have been important to this evaluation.

[1] Grinsztajn et al. "Why do tree-based models still outperform deep learning on typical tabular data?." NeurIPS (2022)

[2] Shwartz-Ziv et al. "Tabular data: Deep learning is not all you need." Information Fusion 81 (2022)

## Technical Quality and Correctness
The paper is clearly written and follows a logical flow. Experiments are solid and extensive, evaluations are also evaluated properly with many considerations such as different evaluations of performance (performance score, ranking) and efficiency. I also appreciate the consideration of statistical significance of the results.
Two critiques on the general quality of the paper are:
- Figures 2 and 3 feel crowded and a little difficult to follow from the descriptions. Maybe splitting these figures and also a better description could help.
- The paper has some minor writing errors which could be fixed (e.g. Abstract (hyperparameters), Figure 2 description (doubled for each), Section 4 (explores))

## Contributions and Review Summary
Overall , this paper provides a systematic evaluation of pipeline combinations for tabular classification tasks. Constructing pipelines that combine preprocessing techniques with various models is a fundamental aspect of many machine learning systems, and this work offers an evaluation of such combinations. However, significant concerns limit its contribution. The experiments mainly reaffirm well-established findings, such as the superiority of ensemble tree-based methods, regardless of preprocessing techniques. While this work is competently executed and may serve as a useful reference point, the above stated weaknesses collectively reduce its impact.

## Comments to Improve Paper
- See Technical Quality + Correctness + Weaknesses

**Review Confidence:**

3

**Review Rating:**

4

---

### Official Review · Reviewer_JHZg · 2025-04-30

**Comments To Authors:**

***Summary Of Contributions***

The authors presented a benchmark that measured the effectiveness of different machine learning pipelines.

***Potential Impact On The Field Of AutoML***

First, the work shows limited novelty and does not introduce a new idea.

Second, the contribution in terms of empirical evaluation appears to be marginal. This work only considers basic machine learning methods available in the scikit-learn package, and omits well-known and well-performing methods such as xgboost, lighgbm and catboost.

Consequently, the impact of this work appears to be limited.


***Technical Quality And Correctness***

In the proposed experiment prepared classification datasets (211) from different sources, it would be worth considering the use of already existing and accepted datasets from known benchmarks.
The experiment space looks extensive, but does not cover all aspects - for example, models outside scikit-learn were missing.
In describing the models used, the authors point to the extra tress model as a tree-based algorithm, which can be properly described as ensemble-based.
The paper considers 500 different configurations, and does not state what the dimension of all possible configurations is and what percentage was covered by the study.
A big problem with the results may be the limitation of 10GB of RAM for more than 1 million pipelines.
The analysis of the results regarding the experiment is not clear in many places. This is caused by repeated claims that something is the best.
The experiments used models with random hyperparameters. One concern is that the model hyperparameters may be different in each pipeline, it would be better to use grid search or use drawn values for all models.


Page 1, 3rd paragraph: steges -> stages.

The charts contain incorrect descriptions relative to the posted text in the article. In addition, showing min/max values and standard deviation in a different order makes it difficult to compare results.

Figure 5 - no units of time.


***Overall Review***

*Positive*: This paper considers an important and practical topic of machine learning pipelines.

*Negative*: Only simple methods are considered in work. No novel idea is proposed. There are concerns about the experiment design. The presentation of this work is unclear at several places. Overall, the contribution seems to be marginal, providing limited practical guidance for practitioners.

**Review Confidence:**

5

**Review Rating:**

3

---

### Review · Reproducibility_Reviewer_vFED · 2025-04-30

**Comments To Authors:**

The code repository would benefit from improved documentation and usability. While the main functionality is present, some important steps are missing or unclear, which makes reproducing the results more difficult than necessary.

The authors mention that the experiments are based on OpenML datasets and provide the corresponding dataset IDs. However, the datasets are not included in the repository. The code expects these datasets to be available locally in a specific pickle format, but there are no instructions or scripts to help users download and prepare the data in this format.

I was able to manually prepare a few datasets, however, this process would be much easier if the repository included a script to automatically download and format datasets from OpenML using their IDs. If that’s not feasible, the authors should at least provide clear instructions for preparing the datasets.

I also tried to run experiments using run.py as described in Step 5 of the instructions:
python3.10 run.py <result_path> <dataset_path> <seed>

However, this command fails to mention the required number of components argument, which is necessary for the script to execute.

When I ran run.py on OpenML dataset with ID 6  with 500 configurations and seed 0 (as described in the experiments), I was not able to reproduce the reported results. The experiments across all folds and configurations has status.CRASHED in the status column. I wasn’t able to identify the cause.

On a positive note, the authors provide a precomputed pipeline experiments database, and once I downloaded it, I was able to successfully replicate the analysis portion of the study.

**Review Confidence:**

4

**Review Rating:**

4

---

### Official Review · Reviewer_k23k · 2025-05-01

**Comments To Authors:**

**A summary of  main contributions**

This paper presents a large-scale benchmark that explores the performance of 1 million different machine learning pipelines and focuses on the interaction between feature preprocessing techniques and classifiers. The study aims to an unbiased evaluation of pipeline composition by using a random search strategy. The study uses 16 classifications algorithms and 13 feature preprocessing techniques for various strategies.

**Potential impact on the field of AutoML**

Potential medium to high impact in AutoML field. The large scale empirical data is valuable for AutoML research.

**Relation to previous work**

The authors acknowledge the existing research in hyperparameter tuning of pipeline optimization and highlight the different focus of their work on the interaction between the classifiers and preprocessors.

**Strength**

- The scale and the scope of the benchmark is impressive and diverse. Extensive coverage provides good statistical power to the findings.

- Rather than focusing on isolated components within pipeline it focuses on the interaction which is novel.

- Using random sampling ensures unbiased exploration making it more generalized.

**Weaknesses**

- The impact of resource constraint is acknowledged by the authors.

-  The authors acknowledge that hyperparameters were not exhaustively explored, instead relying on predefined ranges from Auto-sklearn's configuration space. It is not clear to me if this can cause the true potential of some algorithms not reflected well.

**Review Confidence:**

3

**Review Rating:**

8

---

### Meta-Review · Area_Chair_VayA · 2025-05-05

**Recommendation:** Accept
**Confidence:** 4

**Metareview:**

The paper empirically studies the combination and interaction of pre-processing and classification methods to guide practitioners and AutoML system developers. The main findings are that tree-based ensemble methods can perform well without pre-processing and hyperparameter tuning, whereas kernel methods are less competitive given the same budget. Reviewers mostly agree that the empirical evaluation is extensive and the meta-dataset could be a helpful reference for future research on meta-learning.

Limitations that were raised:
  * Model coverage. The absence of recent algorithms, such as XBBoost and deep learning models, limits relevance.
  * Search strategy. Random search could limit the significance of the findings since more sophisticated methods could improve results for some models. However, the reviewer disagrees on what would've been a better choice.
  * Novelty. The findings mostly reaffirm prior findings, limiting novelty.
  * Reproducibility. There were concerns regarding reproducibility and documentation of the generated results (which I expect are easy to fix)

Considering the mentioned strengths and weaknesses and the review criteria for the “(D)ataset” submission category, I believe that the proposed meta-dataset is a valuable resource. Thus, I recommend **accept for the main track**, encouraging the authors to address/discuss these limitations and issues in a potential final version of their work.